# Differentially Private Graph Learning via Sensitivity-Bounded Personalized PageRank

**Alessandro Epasto**
Google Research
aepasto@google.com

**Vahab Mirrokni**
Google Research
mirrokni@google.com

**Bryan Perozzi**
Google Research
bperozzi@google.com

**Anton Tsitsulin**
Google Research
tsitsulin@google.com

**Peilin Zhong**
Google Research
peilinz@google.com

## Abstract

Personalized PageRank (PPR) is a fundamental tool in unsupervised learning of graph representations such as node ranking, labeling, and graph embedding. However, while data privacy is one of the most important recent concerns, existing PPR algorithms are not designed to protect user privacy. PPR is highly sensitive to the input graph edges: the difference of only one edge may cause a large change in the PPR vector, potentially leaking private user data.

In this work, we propose an algorithm which outputs an approximate PPR and has provably bounded sensitivity to input edges. In addition, we prove that our algorithm achieves similar accuracy to non-private algorithms when the input graph has large degrees. Our sensitivity-bounded PPR directly implies private algorithms for several tools of graph learning, such as, differentially private (DP) PPR ranking, DP node classification, and DP node embedding. To complement our theoretical analysis, we also empirically verify the practical performances of our algorithms.

## 1 Introduction

Personalized PageRank (PPR) [15], has been a workhorse of graph mining and learning for the past twenty years. Given a graph $G$, and a source node $s$, the PPR vector of node $s$ defines a notion of proximity of the other nodes in the graph to it. More precisely, the proximity of $s$ to $v$, is defined by the probability that a biased random walk starting in $s$, visits node $v$.

This elegant variation of the celebrated PageRank algorithm [24] has found widespread use in different application areas of computer science, including web search [7], link prediction [22], network analysis [17, 13], graph clustering [3], natural language processing [27], spam and fake account detection [1, 2]. More recently, PPR has also been used in graph neural networks [21] and graph representation learning [26] including to speed up the computation of graph-based learning algorithms [5, 6].

Despite the widespread use of PPR, and the extensive algorithmic literature dedicated to its efficient approximation [4, 3, 12, 16], to the best of our knowledge no prior work has attempted to compute PPR vectors in a privacy-preserving manner.

In this work, we address this limitation by defining the first approximate method for PPR computation with differential privacy [8]. We focus on a standard notion of differential privacy (DP) for graphs known as edge-level DP. In this notion, two unweighted, undirected graphs $G = (V, E)$ and $G' = (V, E')$, are deemed neighbors if they differ only in the presence of a single edge [23, 10]. An algorithm $\mathcal{A}$ is then said to be edge-level $\epsilon$-differentially private if the difference in the probability of observing any particular outcome from the algorithm when run on $G$ vs $G'$ is bounded: $\Pr[\mathcal{A}(G) \in S] \le e^\epsilon \cdot \Pr[\mathcal{A}(G') \in S]$.

36th Conference on Neural Information Processing Systems (NeurIPS 2022).

Edge-level DP guarantees a strong notion of plausible deniability for the existence of an edge in the graph. This is especially critical in graph-based learning applications, where nodes correspond to humans, and edges depend on personal relationships, which can be highly private and sensitive. Achieving edge-level DP ensures that an attacker observing the output of the algorithm is information-theoretically bounded in their ability to uncover any specific user pair connection.

On top of this notion, we also explore a popular variation of differential privacy used in personalization [18] that follows the concept of joint differential privacy [20]. More precisely we provide Personalized PageRank algorithms that are joint, edge-level differentially-private with respect to the neighborhood of the source node $s$. This means that we can provide the user corresponding to $s$, with an approximate Personalized PageRank of $s$ that depends on the edges incident to $s$ but that protects the information on edges of the rest of the graph. This notion is especially relevant in the context of personalization of results in social networks using PPR, where user data can be safely used to provide an output to the user, but must be protected from leaking to others.

## 1.1 Our results and outline of the paper

Differential privacy forces an algorithm to be insensitive to changes of an edge in the graph. This makes the design of DP PPR algorithms especially challenging as a single edge removal or addition may result in dramatically different PPR vectors, thus potentially exposing private user data. Our first contribution is to propose an algorithm that approximates the PPR vector with a provably bounded sensitivity to edge changes. This technical contribution directly leads to the design of edge-level DP (and joint edge-level DP) algorithms for computing approximate PPR vectors. We believe that this technique may have broader applications in the design of DP graph algorithms.

From a theoretical standpoint, we prove that our private algorithms achieve similar accuracy as non-private approximation algorithms when the input graph has a large enough minimum degree (while the privacy guarantee holds for all graphs of any degree). This dependency on the degree is theoretically justified as we show non-trivial approximation requires large enough degrees.

The main ingredient of our DP algorithms is a novel (non-private) sensitivity-bounded approximate PPR algorithm (Algorithm 2). For any input parameter $0 < \sigma < 1$, the sensitivity of the output of the algorithm in the (joint) edge-level DP case is always upper bounded by $\sigma$. In addition, we show that the algorithm has an $O(\sigma)$ additive error to the ground truth PPR when the minimum degree of the graph is $\Omega(1/\sigma)$ in the DP case (resp. $\Omega(\sqrt{1/\sigma})$ in the joint-DP case). We show that this requirement on the minimum degree for approximation guarantees is almost tight due to hard instances that we present in Appendix A. We then use our sensitivity-bounded algorithm to obtain an edge-level DP (resp. joint edge-level DP) algorithm that, for a graph of minimum degree at least $D$, has $O(1/D)$ additive error (resp. $O(1/D^2)$ error). Then, we focus on applications of differentially-private PPR including computing graph embeddings. We provide provably edge-level DP and joint edge-level DP graph embedding algorithm in Section 5. Finally, in Section 6, we empirically evaluate the performance of our differentially private PPR rankings, as well as that of the embedding methods we design, in several down-stream graph-learning tasks such as node ranking and classification.

To the best of our knowledge, our paper presents the first approximation algorithm with theoretical guarantees for differentially private PPR. This result contributes to the still quite short list of private graph algorithms with provable approximation guarantees that have been developed so far [23, 10, 11, 31, 29],

## 2 Preliminaries

We consider an undirected and unweighted graph $G$ with node set $V = \{v_1, v_2, \cdots, v_n\}$ and edge set $E$. Let $A$ be the adjacency matrix where $A_{i,j} = 1$ indicates an edge between $v_i$ and $v_j$, and $A_{i,j} = 0$ otherwise. Let $\Lambda$ be the diagonal matrix where $\Lambda_{i,i} = d(v_i)$ denotes the degree of $v_i$. We use $\mathbf{0}^n$ to denote an $n$-dimensional all-zero vector and use $e_i \in \mathbb{R}^n$ to denote the one-hot vector where the $i$-th entry is 1 and other entries are 0. Let $[n]$ denote the set $\{1, 2, \cdots, n\}$. When there is no ambiguity, we sometimes abuse the notation between $[n]$ and $V$, i.e., using $v \in [n]$ to denote a node or $i \in V$ denotes an index between 1 and $n$. For $x \in \mathbb{R}^k$, we denote $\|x\|_1 = \sum_{i \in [k]} |x_i|$ and $\|x\|_\infty = \max_{i \in [k]} |x_i|$. We use $\mathrm{Lap}(b)$ ($b > 0$) to denote the Laplace distribution with density function $f(x) = \frac{1}{2b} \cdot \exp(-|x|/b)$. The cumulative distribution function of $\mathrm{Lap}(b)$ is $F(x) = \begin{cases} \frac{1}{2} \cdot \exp(x/b), & x < 0, \\ 1 - \frac{1}{2} \cdot \exp(-x/b), & x \geq 0. \end{cases}$. We will use the following fact in our analysis.

**Fact 2.1.** *Consider $Y$ drawn from* $\mathrm{Lap}(b)$. *For any $\delta \in (0, 1)$,* $\Pr[|Y| > b \ln(1/\delta)] = \delta$.

## 2.1 Personalized PageRank

Personalized PageRank (PPR) takes an input distribution of starting nodes $s \in \mathbb{R}^n$, and starts a lazy random walk with teleport probability $\alpha \in (0,1)$. Typically, $s = e_i$ for some $i \in [n]$, which enforces the random walk starting from the source node $v_i$. If there is no ambiguity, we abuse the notation to denote with $s$ the source node. The output PPR vector is the stationary distribution of the random walk. Precisely, let $W = \frac{1}{2}(I + \Lambda^{-1}A)$ be the lazy random walk transition matrix.[1] The PPR vector $\mathbf{p}(s)$ is defined recursively as: $\mathbf{p}(s) = \alpha \cdot s + (1 - \alpha) \cdot \mathbf{p}(s) \cdot W$. In many applications, it is good enough to use approximate PPR vectors.

**Definition 2.2** ($\xi$-approximate PPR, see e.g., [3]). *For $\xi > 0$, a $\xi$-approximate PPR vector for $\mathbf{p}(s)$ is a PPR vector $\mathbf{p}(s - r)$ where $r$ is a non-negative $n$-dimensional vector called residual vector and $\|r\|_\infty \leq \xi$.*

We also study the following natural variant of the approximate PPR.

**Definition 2.3** (($\xi, \eta$)-approximate PPR). *For $\xi, \eta > 0$, a $(\xi, \eta)$-approximate PPR vector $p$ for $\mathbf{p}(s)$ satisfies $\|p - \mathbf{p}(s - r)\|_\infty \leq \eta$ where $r$ is a non-negative $n$-dimensional vector and $\|r\|_\infty \leq \xi$.*

Note that $\xi$ denotes the error introduced by the residual and $\eta$ denotes the error introduced by the PPR vector itself. Two types of errors are well studied in the literature: for residual error $\xi$, see e.g., [3]; for PPR vector error $\eta$, see e.g., [16].

## 2.2 Differential Privacy

We consider edge-level DP and joint edge-level DP for graph algorithms. Given a graph $G$, we denote $\Gamma(G)$ as the set of all neighboring graphs of $G$, i.e., $\forall G' \in \Gamma(G)$, $G'$ can be obtained from $G$ by either addition or removal of an edge.

**Definition 2.4** (Edge-level DP [10] and joint edge-level DP [20]). *A randomized graph algorithm $\mathcal{A}$ is edge-level $\varepsilon$-DP if for any input graphs $G, G'$ satisfying $G' \in \Gamma(G)$, and for any subset $S$ of possible outputs of $\mathcal{A}$, it holds $\Pr[\mathcal{A}(G) \in S] \leq e^\varepsilon \cdot \Pr[\mathcal{A}(G') \in S]$. Let $V$ be the set of $n$ nodes (users). For joint edge-level DP we assume that a family of $n$ (personalized) graph algorithms $\mathcal{A} = (\mathcal{A}_1, \mathcal{A}_2, \cdots, \mathcal{A}_n)$ is run on the graph and the output of $\mathcal{A}_i$ is provided only to user (node) $v_i \in V$. The family $\mathcal{A}$ is joint edge-level $\varepsilon$-DP, if for any $x, y \in V$ and for any two neighboring graphs $G, G'$ that only differ on edge $(x, y)$, for any $v \neq x, y$ and for any subset $S$ of possible outputs of $\mathcal{A}_v$, it always holds $\Pr[\mathcal{A}_v(G) \in S] \leq e^\varepsilon \cdot \Pr[\mathcal{A}_v(G') \in S]$.[2]*

For $s \in V$, let us denote $\Gamma_s(G)$ as the set of graphs $G' \in \Gamma(G)$ satisfying that $G'$ and $G$ differ on an edge that is *not* incident to $s$. It is easy to verify that the joint edge-level $\varepsilon$-DP is equivalent to asking for $\forall s \in V$, $\forall$ subset $S$ of possible outputs of $\mathcal{A}_s$, $\forall G, G' \in \Gamma_s(G)$, $\Pr[\mathcal{A}_s(G) \in S] \leq e^\varepsilon \cdot \Pr[\mathcal{A}_s(G') \in S]$. In the remainder of the paper, we will simply call edge-level $\varepsilon$-DP as $\varepsilon$-DP and joint edge-level $\varepsilon$-DP as joint $\varepsilon$-DP.

In the following, we will briefly review several other related definitions and theorems for DP.

**Definition 2.5** (Sensitivity [8]). *Consider a function $f$ whose input is a graph and whose output is in $\mathbb{R}^k$. The sensitivity $S_f$ is defined as $S_f = \max_{G, G': G' \in \Gamma(G)} \|f(G) - f(G')\|_1$. Consider a family $\mathcal{F}$ of functions $f_1, f_2, \cdots, f_n$ where each takes a graph as input and outputs a vector in $\mathbb{R}^k$. The joint sensitivity $S_\mathcal{F}$ is defined as $S_\mathcal{F} = \max_{s \in [n], G, G' \in \Gamma_s(G)} \|f_s(G) - f_s(G')\|_1$.*

Moreover, when this simplifies the presentation, we will refer to $\epsilon$-DP and sensitivity as *non-joint* DP and *non-joint* sensitivity to oppose them to *joint* DP and *joint* sensitivity.

**Theorem 2.6** (Laplace mechanism [8]). *Consider a function $f$ whose input is a graph and whose output is in $\mathbb{R}^k$. Suppose $f$ has sensitivity $S_f$. Then the algorithm $\mathcal{A}(G)$ which outputs $f(G) + (Y_1, Y_2, \cdots, Y_k)$ is $\varepsilon$-DP where $Y_i$ are independent $\mathrm{Lap}(S_f/\varepsilon)$ random variables. Similarly, consider*

---

[1]This is equivalent to the standard random walk matrix up to a change in $\alpha$ (see [3]). We use the lazy walk for consistency with prior work.

[2]Note this is slightly weaker than the original definition of joint DP which asks for $\forall$ subsets $S_1, S_2, \cdots, S_n$ of possible outputs of $\mathcal{A}_1, \mathcal{A}_2, \cdots, \mathcal{A}_n$ respectively, $\Pr[\mathcal{A}_{-x,-y}(G) \in S_{-x,-y}] \leq e^\varepsilon \cdot \Pr[\mathcal{A}_{-x,-y}(G') \in S_{-x,-y}]$, where $\mathcal{A}_{-x,-y}$ is the tuple of $n - 2$ outputs of $\mathcal{A}_1, \mathcal{A}_2, \cdots, \mathcal{A}_n$ except $\mathcal{A}_x, \mathcal{A}_y$, and $S_{-x,-y}$ is the cartesian product of $S_1, S_2, \cdots, S_n$ except $S_x, S_y$.

Our definition protects the privacy of each user, however, as we assume that output of $\mathcal{A}_i$ is available to (node) $v_i \in V$ only.

*a family $\mathcal{F}$ of functions $f_1, f_2, \cdots, f_n$ with joint sensitivity $S_\mathcal{F}$. Then the family of $\mathcal{A}_1, \mathcal{A}_2, \cdots, \mathcal{A}_n$ is joint $\varepsilon$-DP where $\mathcal{A}_i(G)$ outputs $f_i(G) + (Y_{i,1}, Y_{i,2}, \cdots, Y_{i,k})$, and $Y_{i,j}$ are independent $\mathrm{Lap}(S_\mathcal{F}/\varepsilon)$ random variables.*

**Theorem 2.7** (Composition [9])**.** *Consider two algorithms $\mathcal{A}_1 : \mathcal{X} \to \mathcal{Y}, \mathcal{A}_2 : \mathcal{Y} \times \mathcal{X} \to \mathcal{Z}$. Suppose $\mathcal{A}_1(\cdot)$ is $\varepsilon_1$-DP, and $\mathcal{A}_2(Y, \cdot)$ is $\varepsilon_2$-DP for any given $Y \in \mathcal{Y}$. Then the algorithm $\mathcal{A}_3 : \mathcal{X} \to \mathcal{Z}$ which is defined as $\mathcal{A}_3(X) = \mathcal{A}_2(\mathcal{A}_1(X), X)$ is $(\varepsilon_1 + \varepsilon_2)$-DP.*

## 3 Warm-Up: Push-Flow on Graphs with High Degrees

As a warm-up, let us start with a standard push-flow algorithm for PPR [3] and provide a novel analysis for bounding the sensitivity when each node has degree at least $D$. The non-private push-flow algorithm is described in Algorithm 1.[3]

---

**Algorithm 1** PUSHFLOW$(G, s, \alpha, \xi)$

---

1: **Input:** Graph $G = (V, E)$, source node $s \in V$, teleport probability $\alpha$, precision $\xi$.
2: **Output:** Approximate PPR vector for $\mathbf{p}(s)$.
3: Initialize $S^{(0)} \leftarrow \{s\}, p^{(0)} \leftarrow \mathbf{0}^n, r^{(0)} \leftarrow e_s$, and $R \leftarrow \lceil \ln(1/\xi)/\alpha \rceil$.
4: **for** $i := 1 \to R$ **do**
5:     Let $S^{(i)} \leftarrow S^{(i-1)}$. Let $p^{(i)}, r^{(i)} \leftarrow \mathbf{0}^n$.
6:     **for** Each node $v \in S^{(i-1)}$ **do**
7:         $p_v^{(i)} \leftarrow p_v^{(i-1)} + \alpha \cdot r_v^{(i-1)}, r_v^{(i)} \leftarrow r_v^{(i)} + (1-\alpha)/2 \cdot r_v^{(i-1)}$
8:         For each neighbor $u$, i.e., $(v, u) \in E$: $r_u^{(i)} \leftarrow r_u^{(i)} + (1-\alpha)/2 \cdot r_v^{(i-1)}/d(v), S^{(i)} \leftarrow S^{(i)} \cup \{u\}$.
9:     **end for**
10: **end for**
11: Output $p^{(R)}$.

---

**Lemma 3.1.** *Algorithm 1 outputs a $\xi$-approximate PPR vector in $O(|E| \log(1/\xi)/\alpha)$ time.*

The proof of Lemma 3.1 follows the analysis idea of [3]. For completeness, we put the proof in Appendix B. Next, we prove the sensitivity of Algorithm 1 when every node has degree at least $D$.

**Theorem 3.2** (Sensitivity of PUSHFLOW)**.** *Consider two graphs $G = (V, E), G' = (V, E')$ where $G' \in \Gamma(G)$. In addition, both $G$ and $G'$ have a minimum degree at least $D$. Let $p, p'$ be the output of PUSHFLOW$(G, s, \alpha, \xi)$ and PUSHFLOW$(G', s, \alpha, \xi)$ respectively. Then if $G' \in \Gamma_s(G)$, $\|p - p'\|_1 \leq \frac{2 \cdot (1-\alpha)}{\alpha \cdot D^2}$. Otherwise, $\|p - p'\|_1 \leq \frac{2 \cdot (1-\alpha)}{\alpha \cdot D}$.*

*Proof.* Without loss of generality, suppose $G'$ has one more edge $(x, y)$ than $G$, i.e., $E' = E \cup \{(x, y)\}$. Let $p^{(i)}, r^{(i)}$ be the same as described in Algorithm 1 when running PUSHFLOW$(G, s, \alpha, \xi)$. Similarly, let $p'^{(i)}, r'^{(i)}$ be the vectors $p^{(i)}, r^{(i)}$ described in Algorithm 1 when running PUSHFLOW$(G', s, \alpha, \xi)$. Thus, our goal is to bound $\|p^{(R)} - p'^{(R)}\|_1$. It suffices to bound $\|p^{(R)} - p'^{(R)}\|_1 + \|r^{(R)} - r'^{(R)}\|_1$. Let $d(v)$ denote the degree of $v$ in $G$ and let $d'(v)$ denote the degree of $v$ in $G'$. Consider $i \in [R]$. We have:

$$\|p^{(i)} - p'^{(i)}\|_1 + \|r^{(i)} - r'^{(i)}\|_1 \leq \sum_{v \in V} \left| (p_v^{(i-1)} + \alpha \cdot r_v^{(i-1)}) - (p_v'^{(i-1)} + \alpha \cdot r_v'^{(i-1)}) \right| + \sum_{v \in V} \frac{1-\alpha}{2} \cdot \left| r_v^{(i-1)} - r_v'^{(i-1)} \right|$$

$$+ \sum_{v \in V} \frac{1-\alpha}{2} \cdot \left| \sum_{u:(u,v) \in E} \frac{r_u^{(i-1)}}{d(u)} - \sum_{u':(u',v) \in E'} \frac{r_{u'}'^{(i-1)}}{d'(u')} \right|$$

$$\leq \|p^{(i-1)} - p'^{(i-1)}\|_1 + \frac{1+\alpha}{2} \cdot \|r^{(i-1)} - r'^{(i-1)}\|_1 \tag{1}$$

$$+ \frac{1-\alpha}{2} \cdot \left( \sum_{v \in V} \sum_{u:(u,v) \in E} \left| \frac{r_u^{(i-1)}}{d(u)} - \frac{r_u'^{(i-1)}}{d'(u)} \right| + \left( \frac{r_x'^{(i-1)}}{d'(x)} + \frac{r_y'^{(i-1)}}{d'(y)} \right) \right) . \tag{2}$$

---

[3]The algorithm described in Algorithm 1 is slightly different from the original push-flow algorithm of [3]. In each iteration, instead of pushing flow for the node with the largest residual, we push flows for all nodes that were visited. This variant gives us a better bound of sensitivity.

By reordering the terms, part (2) is equal to:

$$\frac{1-\alpha}{2} \cdot \sum_{u \in V \setminus \{x,y\}} d(u) \cdot \left| \frac{r_u^{(i-1)}}{d(u)} - \frac{r_u'^{(i-1)}}{d(u)} \right|$$

$$+ \frac{1-\alpha}{2} \cdot \left( \left( d(x) \cdot \left| \frac{r_x^{(i-1)}}{d(x)} - \frac{r_x'^{(i-1)}}{d(x)+1} \right| + \frac{r_x'^{(i-1)}}{d(x)+1} \right) + \left( d(y) \cdot \left| \frac{r_y^{(i-1)}}{d(y)} - \frac{r_y'^{(i-1)}}{d(y)+1} \right| + \frac{r_y'^{(i-1)}}{d(y)+1} \right) \right) \cdot$$

Notice that $d(x) \cdot \left| \frac{r_x^{(i-1)}}{d(x)} - \frac{r_x'^{(i-1)}}{d(x)+1} \right| + \frac{r_x'^{(i-1)}}{d(x)+1} \leq |r_x^{(i-1)} - r_x'^{(i-1)}| + \frac{2 \cdot r_x'^{(i-1)}}{d(x)+1}$. Similar arguments hold for $y$. Thus, part (2) is at most $(1-\alpha)/2 \cdot (\|r^{(i-1)} - r'^{(i-1)}\|_1 + 2 \cdot (r_x'^{(i-1)}/d'(x) + r_y'^{(i-1)}/d'(y)))$. Therefore (1)+(2)$\leq \|p^{(i-1)} - p'^{(i-1)}\|_1 + \|r^{(i-1)} - r'^{(i-1)}\|_1 + (r_x'^{(i-1)}/d'(x) + r_y'^{(i-1)}/d'(y))$. Since $d'(x), d'(y) \geq D$, we have: $\|p^{(i)} - p'^{(i)}\|_1 + \|r^{(i)} - r'^{(i)}\|_1 \leq \|p^{(i-1)} - p'^{(i-1)}\|_1 + \|r^{(i-1)} - r'^{(i-1)}\|_1 + (1-\alpha) \cdot \left( r_x'^{(i-1)} + r_y'^{(i-1)} \right) /D$.

In Appendix C, we show that $r_x'^{(i-1)}, r_y'^{(i-1)} \leq (1-\alpha)^{i-1}$ and if in addition $s \neq x, y$, $r_x'^{(i-1)}, r_y'^{(i-1)} \leq (1-\alpha)^{i-1}/D$. Thus, if $s \neq x, y$, we have: $\|p^{(R)} - p'^{(R)}\|_1 \leq 2 \cdot (1-\alpha)/D \cdot \sum_{i=1}^{R}(1-\alpha)^{i-1}/D \leq \frac{2 \cdot (1-\alpha)}{\alpha \cdot D^2}$. Otherwise, we have $\|p^{(R)} - p'^{(R)}\|_1 \leq 2 \cdot (1-\alpha)/D \cdot \sum_{i=1}^{R}(1-\alpha)^{i-1} \leq \frac{2 \cdot (1-\alpha)}{\alpha \cdot D}$ $\square$

In Appendix A, we show that the analysis in Theorem 3.2 is tight as the sensitivity of the ground truth PPR in graphs with minimum degree $D$ can be $\Omega(1/D)$ (or $\Omega(1/D^2)$ for joint sensitivity).

If input graphs were always guaranteed to have minimum degree at least $D$, we could obtain a DP or a joint DP PPR algorithm by applying the Laplace mechanism on the output of Algorithm 1 (see Appendix D). However, in practice the input graphs can have low degree nodes. In this case, the sensitivity of the vanilla push-flow algorithm (Algorithm 1) can be very high. In the next section, we address the question of how to modify the algorithm to ensure low sensitivity for *any* input graph.

## 4  Push-Flow with Bounded Sensitivity in General Graphs

In this section, we propose a variant of the push-flow algorithm of which sensitivity (resp. joint sensitivity) is always bounded by an input parameter $\sigma$. As a result, we can apply Laplace mechanism (Theorem 2.6) to obtain a DP PPR (resp. a joint DP PPR) algorithm for all possible general input graphs and thus the added noise is controlled by $\sigma$. In addition, our new algorithm achieves the same approximation as Algorithm 1 when every node in the input graph has a relatively high degree.

We explain the intuition of our algorithm as the following. Recall the analysis of the sensitivity of Algorithm 1 (the proof of Theorem 3.2). The reason that it may introduce a large sensitivity is because every node $x$ with residual $r_x'^{(i-1)}$ may push $\sim r_x'^{(i-1)}/d'(x)$ amount of flow along each of its incident edges. If $d'(x)$ is small, the sensitivity introduced by an edge incident to $x$ can be very large. Therefore, to control the sensitivity, a natural idea is to set a threshold for each edge such that the total pushed flow along each edge can never be above the threshold. We present our sensitivity-bounded push-flow algorithm in Algorithm 2. For sake of presentation, this algorithm and the following ones have a parameter *type* $\in \{joint, non\text{-}joint\}$ indicating whether we are working in the *joint* DP case or in the *non-joint* DP case.

Firstly, we show that the joint/non-joint sensitivity of PUSHFLOWCAP($G, s, \alpha, \xi, \sigma, joint/non\text{-}joint$) is indeed bounded by $\sigma$. We will use the following observation.

**Observation 4.1.** *Consider $p^{(i)}, h^{(i)}, f^{(i)}$ in* PUSHFLOWCAP($G, s, \alpha, \xi, \sigma, joint/non\text{-}joint$). *Then* $\forall v \in V, i \in [R]$: *(1)* $h_v^{(i)} = \sum_{j=1}^{i} f_v^{(j)}$. *(2)* $h_v^{(i)} = \min(\sum_{j=0}^{i-1} r_v^{(j)}, d(v) \cdot T_v)$. *(3)* $p_v^{(i)} = \alpha \cdot h_v^{(i)}$.

Using this observation we show the following lemma on $h_v^{(i)}$.

**Lemma 4.2.** *Consider $h^{(i)}$ and $r^{(i)}$ in* PUSHFLOWCAP($G, s, \alpha, \xi, \sigma, joint/non\text{-}joint$). *$\forall v \in V, i \in [R]$, $h_v^{(i)} = \min \left( r_v^{(0)} + \frac{1-\alpha}{2} \cdot \left( h_v^{(i-1)} + \sum_{u:(u,v) \in E} \frac{h_u^{(i-1)}}{d(u)} \right), d(v) \cdot T_v \right)$.*

**Algorithm 2** PUSHFLOWCAP($G, s, \alpha, \xi, \sigma, \textit{type}$)

---

1: **Input:** Graph $G = (V, E)$, source node $s \in V$, teleport probability $\alpha$, precision $\xi$, sensitivity parameter $\sigma$, and $\textit{type} \in \{\textit{joint}, \textit{non-joint}\}$ indicating whether joint DP sensitivity or (vanilla) DP sensitivity is required.
2: **Output:** Approximate PPR vector for $\mathbf{p}(s)$.
3: Initialize the set of nodes with positive residual $S^{(0)} \leftarrow \{s\}$, PPR $p^{(0)} \leftarrow \mathbf{0}^n$, residual $r^{(0)} \leftarrow e_s$, total pushed flow $h^{(0)} \leftarrow \mathbf{0}^n$, number of rounds $R \leftarrow \lceil \ln(1/\xi)/\alpha \rceil$, and thresholds $T \in \mathbb{R}^n$ such that
      1. If $\textit{type} = \textit{joint}$, $T_s \leftarrow \infty$ and $\forall u \neq s, T_u \leftarrow \sigma/(2 \cdot (2 - \alpha))$.
      2. Otherwise, $\forall u \in V, T_u \leftarrow \sigma/(2 \cdot (2 - \alpha))$.
4: **for** $i := 1 \rightarrow R$ **do**
5:   **for** Each node $v \in S^{(i-1)}$ **do**
6:     $f_v^{(i)} \leftarrow \min(r_v^{(i-1)}, d(v) \cdot T_v - h_v^{(i-1)})$.          //Compute the flow to push for node $v$.
7:     $h_v^{(i)} \leftarrow h_v^{(i-1)} + f_v^{(i)}$.          //Update the total pushed flow of node $v$.
8:     $p_v^{(i)} \leftarrow p_v^{(i-1)}, r_v^{(i)} \leftarrow r_v^{(i-1)} - f_v^{(i)}$.
9:   **end for**
10:   $S^{(i)} \leftarrow S^{(i-1)}$.
11:   **for** Each node $v \in S^{(i-1)}$ with $f_v^{(i)} > 0$ **do**
12:     $p_v^{(i)} \leftarrow p_v^{(i)} + \alpha \cdot f_v^{(i)}, r_v^{(i)} \leftarrow r_v^{(i)} + (1-\alpha)/2 \cdot f_v^{(i)}$.      // Do actual flow push
13:     For each neighbor $u$, i.e., $(v, u) \in E : r_u^{(i)} \leftarrow r_u^{(i)} + (1-\alpha)/2 \cdot f_v^{(i)}/d(v), S^{(i)} \leftarrow S^{(i)} \cup \{u\}$.
14:   **end for**
15: **end for**
16: Output $p^{(R)}$.

---

**Theorem 4.3** (Sensitivity and joint sensitivity of PUSHFLOWCAP). *Consider two graphs $G = (V, E), G' = (V, E')$. Let $p, p'$ be the output of PUSHFLOWCAP($G, s, \alpha, \xi, \sigma, \textit{type}$) and PUSHFLOWCAP($G', s, \alpha, \xi, \sigma, \textit{type}$) respectively. For $\textit{type} = \textit{joint}$ and $G' \in \Gamma_s(G)$, or $\textit{type} = \textit{non-joint}$ and $G' \in \Gamma(G)$, then $\|p - p'\|_1 \leq \sigma$.*

*Proof.* Without loss of generality, let $G' \in \Gamma(G)$ has exactly one more edge $(x, y)$ than $G$, i.e., $E' = E \cup \{(x, y)\}$. Let $p^{(i)}, h^{(i)}$ be the same as described in Algorithm 2 when running PUSH-FLOWCAP($G, s, \alpha, \xi, \sigma, \textit{type}$). Similarly, let $p'^{(i)}, h'^{(i)}$ be the vectors $p^{(i)}, h^{(i)}$ described in Algorithm 2 when running PUSHFLOWCAP($G', s, \alpha, \xi, \sigma, \textit{type}$). Let $d(v)$ denote the degree of $v$ in $G$ and let $d'(v)$ denote the degree of $v$ in $G'$. To prove the lemma, our goal is to bound $\|p^{(R)} - p'^{(R)}\|_1$. According to Observation 4.1, we have $\|p^{(R)} - p'^{(R)}\|_1 = \alpha \cdot \|h^{(R)} - h'^{(R)}\|_1$. It suffices to bound $\|h^{(R)} - h'^{(R)}\|_1$. Notice that $\forall a_1, a_2, a_3, a_4 \in \mathbb{R}$, it is easy to verify that $|\min(a_1, a_2) - \min(a_3, a_4)| \leq |a_1 - a_3| + |a_2 - a_4|$. Consider $i \in [R]$. According to Lemma 4.2, for every $v \neq x, y$, we have:

$$|h_v^{(i)} - h_v'^{(i)}| \leq |r_v^{(0)} - r_v'^{(0)}| + \frac{1-\alpha}{2}\left(|h_v^{(i-1)} - h_v'^{(i-1)}| + \sum_{u:(u,v)\in E}\left|\frac{h_u^{(i-1)}}{d(u)} - \frac{h_u'^{(i-1)}}{d'(u)}\right| + \left|(d(v) - d'(v))T_v\right|\right)$$

$$= \frac{1-\alpha}{2}\cdot\left(\left|h_v^{(i-1)} - h_v'^{(i-1)}\right| + \sum_{u:(u,v)\in E}\left|\frac{h_u^{(i-1)}}{d(u)} - \frac{h_u'^{(i-1)}}{d'(u)}\right|\right),$$

where the last step follows from $r_v^{(0)} = r_v'^{(0)}$ and $d(v) = d'(v)$. Suppose $v \in \{x, y\}$. Let $v'$ be the other node in $x, y$, i.e., $v' \in \{x, y\} \setminus \{v\}$. We have:

$$|h_v^{(i)} - h_v'^{(i)}|$$

$$\leq |r_v^{(0)} - r_v'^{(0)}| + \frac{1-\alpha}{2}\cdot\left(|h_v^{(i-1)} - h_v'^{(i-1)}| + \frac{h_{v'}'^{(i-1)}}{d'(v')} + \sum_{u:(u,v)\in E}\left|\frac{h_u^{(i-1)}}{d(u)} - \frac{h_u'^{(i-1)}}{d'(u)}\right|\right) + |(d(v) - d'(v))T_v|$$

$$= \frac{1-\alpha}{2}\cdot\left(\left|h_v^{(i-1)} - h_v'^{(i-1)}\right| + \sum_{u:(u,v)\in E}\left|\frac{h_u^{(i-1)}}{d(u)} - \frac{h_u'^{(i-1)}}{d'(u)}\right|\right) + \frac{1-\alpha}{2}\cdot\frac{h_{v'}'^{(i-1)}}{d'(v')} + T_v,$$

where the last step follows from $|d(v) - d(v')| = 1$ and $r_v^{(0)} = r_v'^{(0)}$. Therefore,

$$\|h^{(i)} - h'^{(i)}\|_1$$

$$\leq \left( \left\| h^{(i-1)} - h'^{(i-1)} \right\|_1 + \sum_{v \in V} \sum_{u:(u,v) \in E} \left| \frac{h_u^{(i-1)}}{d(u)} - \frac{h_u'^{(i-1)}}{d'(u)} \right| + \frac{h_x'^{(i-1)}}{d'(x)} + \frac{h_y'^{(i-1)}}{d'(y)} \right) \cdot \frac{1-\alpha}{2} + T_x + T_y$$

$$= \left( \left\| h^{(i-1)} - h'^{(i-1)} \right\|_1 + \sum_{u \in V \setminus \{x,y\}} \left| h_u^{(i-1)} - h_u'^{(i-1)} \right| \right.$$

$$\left. + d(x) \cdot \left| \frac{h_x^{(i-1)}}{d(x)} - \frac{h_x'^{(i-1)}}{d(x)+1} \right| + d(y) \cdot \left| \frac{h_y^{(i-1)}}{d(y)} - \frac{h_y'^{(i-1)}}{d(y)+1} \right| + \frac{h_x'^{(i-1)}}{d'(x)} + \frac{h_y'^{(i-1)}}{d'(y)} \right) \cdot \frac{1-\alpha}{2} + T_x + T_y.$$

Notice that $d(x) \cdot \left| \frac{h_x^{(i-1)}}{d(x)} - \frac{h_x'^{(i-1)}}{d(x)+1} \right| \leq |h_x^{(i-1)} - h_x'^{(i-1)}| + \frac{h_x'^{(i-1)}}{d(x)+1}$. Similarly, $d(y) \cdot \left| \frac{h_y^{(i-1)}}{d(y)} - \frac{h_y'^{(i-1)}}{d(y)+1} \right| \leq |h_y^{(i-1)} - h_y'^{(i-1)}| + \frac{h_y'^{(i-1)}}{d(y)+1}$. Therefore, we have:

$$\|h^{(i)} - h'^{(i)}\|_1 \leq (1-\alpha) \cdot \left( \left\| h^{(i-1)} - h'^{(i-1)} \right\|_1 + \frac{h_x'^{(i-1)}}{d'(x)} + \frac{h_y'^{(i-1)}}{d'(y)} \right) + T_x + T_y.$$

According to Observation 4.1, we have $\frac{h_x'^{(i-1)}}{d'(x)} \leq T_x$ and $\frac{h_y'^{(i-1)}}{d'(y)} \leq T_y$. Therefore, $\|h^{(i)} - h'^{(i)}\|_1 \leq (1-\alpha) \cdot \|h^{(i-1)} - h'^{(i-1)}\|_1 + (2-\alpha) \cdot (T_x + T_y)$. Since $\|h^{(0)} - h'^{(0)}\|_1 = 0$, we have $\|h^{(R)} - h'^{(R)}\|_1 \leq (2-\alpha) \cdot (T_x + T_y)/\alpha$. Hence, $\|p^{(R)} - p'^{(R)}\|_1 \leq (2-\alpha) \cdot (T_x + T_y)$. If $G' \in \Gamma_s(G)$, i.e., $s \neq x, y$, $T_x + T_y = 2 \cdot \frac{\sigma}{(2 \cdot (2-\alpha))}$. If non-joint sensitivity is considered and $G' \notin \Gamma_s(G)$, i.e., $s \in \{x,y\}$, $T_x + T_y \leq 2 \cdot \frac{\sigma}{(2 \cdot (2-\alpha))}$. We conclude the proof. $\square$

The following lemma shows the running time and the approximation guarantee of Algorithm 2. See Appendix F for the proof.

**Lemma 4.4.** PUSHFLOWCAP$(G, s, \alpha, \xi, \sigma, \text{type})$ *runs in* $O(|E| \log(1/\xi)/\alpha)$ *time. Furthermore, if the minimum degree of $G$ is at least* $\max\left(1/(\alpha \cdot T_s), \sqrt{1/(\alpha \cdot T_u)}\right)$ $(u \neq s)$, *the output of* PUSHFLOWCAP$(G, s, \alpha, \xi, \sigma, \text{type})$ *is exactly the same as the output of* PUSHFLOW$(G, s, \alpha, \xi)$, *i.e., it is a $\xi$-approximate PPR vector for* $\mathbf{p}(s)$.

**Remark 4.5.** *According to the choice of $T_s$ and $T_u$, the above lemma shows that if $\sigma \geq \Omega_\alpha(1/D^2)$ (resp. $\sigma \geq \Omega_\alpha(1/D)$), the output of* PUSHFLOWCAP$(G, s, \alpha, \xi, \sigma, \text{type} = \text{joint})$ *(resp. for type $=$ non-joint) is a $\xi$-approximate PPR vector for* $\mathbf{p}(s)$. *This is near optimal since we show in Appendix A that the joint sensitivity (resp. non-joint sensitivity) of the ground truth PPR of a graph with minimum degree $D$ can be at least $\Omega(1/D^2)$ (resp. $\Omega(1/D)$).*

We apply Laplace mechanism to Algorithm 2 and obtain a DP or joint DP PPR algorithm for general input graphs (see Algorithm 3). We conclude the theoretical guarantees of our algorithm.

---

**Algorithm 3** DPPUSHFLOWCAP$(G, s, \alpha, \xi, \sigma, \varepsilon, \text{type})$

1: **Input:** Graph $G = (V, E)$, source $s \in V$, teleport probability $\alpha$, precision $\xi$, sensitivity parameter $\sigma$, DP parameter $\varepsilon$, and *type* $\in \{\text{joint}, \text{non-joint}\}$ indicating whether joint $\varepsilon$-DP or $\varepsilon$-DP is required.
2: **Output:** $\varepsilon$-DP approximate PPR vector for $\mathbf{p}(s)$.
3: $p \leftarrow$ PUSHFLOWCAP$(G, s, \alpha, \xi, \sigma, \text{type})$.
4: Let $Y_1, Y_2, \cdots, Y_n$ drawn independently from $\text{Lap}\left(\frac{\sigma}{\varepsilon}\right)$
5: Output $p + (Y_1, Y_2, \cdots, Y_n)$.

---

**Corollary 4.6** (Joint DP and DP PPR). *The family of (personalized) algorithms* $\{\mathcal{A}_s(G) := \text{DPPUSHFLOWCAP}(G, s, \alpha, \xi, \sigma, \varepsilon, \text{joint}) \mid s \in V\}$ *is joint $\varepsilon$-DP, and* DPPUSHFLOWCAP$(G, s, \alpha, \xi, \sigma, \varepsilon, \text{non-joint})$ *is $\varepsilon$-DP with respect to $G$ for any $s \in V$. In addition, if the input graph $G$ has a minimum degree at least $D$, the joint $\varepsilon$-DP (resp. $\varepsilon$-DP) output is a $\left(\xi, O_{\alpha,\varepsilon}(\sigma \ln \frac{n}{\delta})\right)$-approximate PPR with probability at least $1 - \delta$ for any $\delta \in (0,1)$ when $\sigma \geq \Omega_\alpha(1/D^2)$ (resp. $\sigma \geq \Omega_\alpha(1/D)$).*

According to above corollary, we want the sensitivity parameter $\sigma$ to be as small as possible since smaller $\sigma$ leads to smaller additive error. On the other hand, too small $\sigma$ may make the approximate PPR obtained by PUSHFLOWCAP (Algortihm 2) deviate from the ground truth PPR. Therefore, if a

graph has a minimum degree $D$, our joint $\varepsilon$-DP PPR (resp. $\varepsilon$-DP PPR) algorithm provides the best theoretical approximation guarantees when $\sigma = \Theta_\alpha(1/D^2)$ (resp. $\sigma = \Theta_\alpha(1/D)$). This matches the lower bound shown in Appendix A: the ground truth PPR has sensitivity $\Omega_\alpha(1/D)$ and joint sensitivity $\Omega_\alpha(1/D^2)$. Thus, our results are actually theoretically optimal up to constant factors. The implication of Corollary 4.6 is hence tight, and it is impossible to have a good theoretical approximation guarantee if $\sigma < o_\alpha(1/D)$ for $\varepsilon$-DP or $\sigma < o_\alpha(1/D^2)$ for joint $\varepsilon$-DP. In the experimental section we show, however, that in practice the algorithm performs well for a vast range of the parameter setting.

## 5 Differentially Private Graph Embeddings

As we discussed before, PPR has plenty of applications in graph learning [21, 26, 6]. In this section, to show the potentiality of our algorithms, we focus on a recent example of the use of PPR for computing graph embedding. We consider InstantEmbedding [26] (see Algorithm 4) which is one practical PPR-based graph embedding algorithm. The algorithm proceeds computing the PPR vector of $s$ and then hashing them to obtain an embedding in $\mathbb{R}^k$ for the node $s$. Using our *DP* PPR algorithm output as input to InstantEmbedding leads trivially to a DP embedding algorithm. However, in this section, we show a better implementation which reduces the amount of noise added using in a slightly more sophisticated way our *sensitivity bounded* PPR algorithm. We think that this technique could be adapted to other uses of PPR.

First we provide a sensitivity bound for the InstantEmbedding algorithm when applying the sensitivity-bounded PPR. As a result, we show how to obtain differentially private InstantEmbedding. The proof of the following theorem (the sensitivity bound) can be found in Appendix H.

---

**Algorithm 4** INSTANTEMBEDDING$(p, k)$

---

1: **Input:** An approximate PPR vector $p$ for $\mathbf{p}(s)$, dimension $k$, and uniform random hash functions $h_k : V \to [k], h_{sgn} : V \to \{-1, 1\}$.
2: **Output:** Embedding vector $w \in \mathbb{R}^k$.
3: Initialize $w \leftarrow \mathbf{0}^k$.
4: **for** $v \in V$ **do**
5: $\quad w_{h_k(v)} \leftarrow w_{h_k(v)} + h_{sgn}(v) \cdot \max(\log(p_v \cdot n), 0)$.
6: **end for**
7: Output $w$.

---

**Theorem 5.1** (Sensitivity-bounded InstantEmbedding)**.** *Consider two neighboring graphs $G = (V, E), G' = (V, E')$ and source node $s$. Let $p, p'$ be approximate PPR vectors for $\mathbf{p}(s)$ with respect to $G$ and $G'$ respectively. Let $w, w'$ be the output of* INSTANTEMBEDDING$(p, k)$ *and* INSTANTEMBEDDING$(p', k)$ *respectively. Let $m$ be the number of non-zero entries of $p - p'$. Then, $\|w - w'\|_1 \le m \cdot \log\left(1 + \|p - p'\|_1 \cdot \frac{n}{m}\right)$.*

Non-zero entries of $p - p'$ in the above theorem can be at most $n$. Thus, $\|w - w'\|_1$ is always at most $\|p - p'\|_1 \cdot n$. If we compute $p \leftarrow$ PUSHFLOWCAP$(G, s, \alpha, \xi, \sigma)$, according to Theorem 4.3, the (joint) sensitivity of $p$ is at most $\sigma$. Then, according to Theorem 5.1, the (joint) sensitivity of output $w$ of Algorithm 4 (using as approximate ppr $p$) is at most $\sigma \cdot n$. This allows us to obtain a (joint) $\varepsilon$-DP version of InstantEmbedding by using the Laplace mechanism (Theorem 2.6) to add $\mathrm{Lap}\left(\sigma \cdot n/\varepsilon\right)$ to each entry of $w$.

## 6 Experiments

In this section, we complement our theoretical analysis with an experimental study of our algorithms in terms of the accuracy of the PPR ranking and node classification. In this experimental section we only consider the joint DP setting due to its practicality in personalization applications typical of PPR and its better performance. Hence all private algorithms reported are joint DP.

**Hyperparameter settings.** Unless otherwise specified, we ran all experiments that use our PUSH-FLOWCAP algorithm (or the non-private PUSHFLOW algorithm [3]) to obtain PPR rankings consistent to the setting commonly used in the literature of $\alpha = 0.15$.[4] Moreover, we set $\xi$ so that the number of iterations $R = 100$ in all algorithms. We use embedding dimensionality $k = 256$. We observe that our algorithm's utility is not strongly affected by the sensitivity parameter $\sigma$. For simplicity, we always set $\sigma = 10^{-6}$, as this parameter generalized across different datasets tested. Detailed experiments on the effects of the sensitivity parameter can be found in the Appendix.

---

[4]The algorithms use the *lazy* random walk [3], so we set $\alpha = 0.08$ to match the non-lazy $\alpha$ of 0.15.

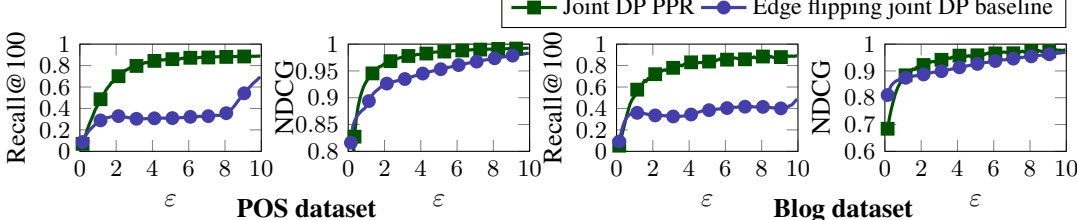

Figure 1: PPR approximation on two datasets.

**Baselines.** To the best of our knowledge, this is the first PPR paper with differential privacy. To compare with relevant joint DP baselines we use the standard randomized response [9] (edge-flipping) baseline applied to the graph. Given the source node $s$, for each (unordered) pair $\{u, v\}$ of nodes $(u, v \neq s)$ we apply the randomized response mechanism on the corresponding entry in the adjacency matrix: with probability $p = 2/(1 + \exp(\epsilon/2))$, the entry is replaced by a uniform at random $\{0, 1\}$, otherwise we keep the true entry. This results in a joint $\epsilon$-DP output over which we run the non-private PPR algorithm [3]. Note that this baseline, for $\epsilon = \Theta(1)$, requires $\Theta(n^2)$ time to generate a DP adjacency matrix, and makes the output matrix dense. This limits the applicability of the baseline mechanism, whereas our approach remains scalable for larger graphs.

We also evaluated another simple joint DP baseline: adding $\text{Lap}\left(\frac{1}{\varepsilon}\right)$ noise[5] to all PPR values obtained by the non-private PUSHFLOW algorithm. We omit these results as they were close to random.

**Datasets.** We experiment on 2 publicly available datasetsavailable from [14, 28]. Table 1 reports basic statistics about these datasets. POS is a word co-occurrence network built from Wikipedia. BlogCatalog a social network of bloggers from the blogcatalog website.

Table 1: Dataset characteristics: number of vertices $|V|$, number of edges $|E|$; number of node labels $|\mathcal{L}|$; average node degree; density defined as $|E|/\binom{|V|}{2}$.

| | **Size** | | | **Statistics** | |
|---|---|---|---|---|---|
| *dataset* | $|V|$ | $|E|$ | $|\mathcal{L}|$ | Avg. deg. | Density |
| POS | 5k | 185k | 40 | 38.7 | $8.1 \times 10^{-3}$ |
| Blogcatalog | 10k | 334k | 39 | 64.8 | $6.3 \times 10^{-3}$ |

### 6.1 PPR Approximation Accuracy

We verify that our algorithms can rank the nodes of real-world graphs in a private way. We randomly sample 1000 nodes and compute "ground-truth" PPR values with the standard power iteration algorithm. Then, we run DPPUSHFLOWCAP algorithm in the joint DP setting. Figure 1 presents the results on the datasets studied. We examine the performance of Joint DP PPR from two standard metrics: Recall@k and normalized discounted cumulative gain (NDCG) [19]. We select Recall@100 and NDCG since the top PPR values are the most important in practical applications of ranking. We observe that the joint DP rankings offer significantly better top-k predictions across datasets and metrics tested than the private baseline.

### 6.2 Node Classification via Private Embeddings

Last, we examine the performance of our joint DP embedding algorithm. We follow the procedure of [25, 14] and evaluate our embeddings on a node classification task in real-world graphs. We report the results in Figure 2. Here, Joint DP and DP represents our respective implementations of Instant-Embedding; DP Baseline + InstantEmbedding represents the obvious baseline of computing Baseline DP PPR followed by the hashing procedure; Non-DP InstantEmbedding is the original non-DP embedding and Random represents a uniform random embedding. Our joint DP algorithm has significantly better performance than the baseline and has non-trivial Micro-F1 even for small $\varepsilon$. In contrast, the baseline does not extract any useful information from the graph in this range for $\varepsilon$ .

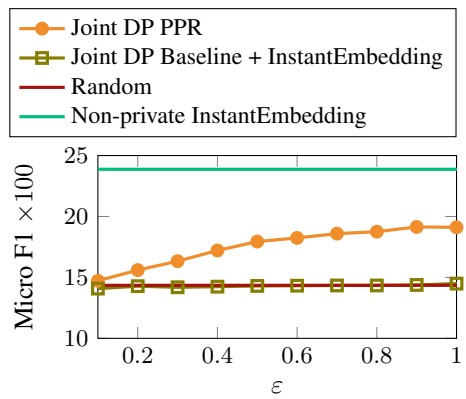

Figure 2: Private embeddings introduced in this paper outperform other competitors and achieve much higher performance on a tight privacy budget.

---

[5]This calibration of noise is needed even in the joint DP setting because of the high sensitivity of PPR.

# 7 Conclusion

In this work, we showed that it is possible to compute approximate PPR rankings with differential privacy. We believe that the techniques developed for bounding the sensitivity of PPR can find applications in other areas of graph-learning. As a future work, we would like to extend the use of our DP PPR algorithm to more machine learning tasks.

**Societal impact and limitations** Our work focuses on developing DP algorithms for data analysis. If used correctly, DP provides strong protection, but has limitations (we refer to standard textbooks on the subject [9]). Moreover, while privacy is a requirement of a responsible computational system, it is not the only one. We encourage reviewing holistically the safety of any application of our work.

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
