# A  Tightness of the Sensitivity Bound for PPR

In this section, we show examples that the sensitivity of a PPR vector of a graph with minimum degree $D$ can be $\Omega(1/D)$ and the sensitivity for joint DP can be $\Omega(1/D^2)$. For simplicity, consider $\alpha = 0.5$. Consider a clique with $D + 1$ nodes. Thus, each node has degree $D$. Let us choose an arbitrary node as the source node. Let $p$ be the PPR vector for $s$. Then it is easy to verify $p_s = \frac{2D+1}{3D+1}$ and $p_v = \frac{1}{3D+1}$ for $v \neq s$.

Suppose we remove the edge between node $s$ and node $x$. Let $p'$ be the new PPR vector for $s$. We can verify that $p'_s = \frac{6D+5}{9D+6}, p'_x = \frac{1}{9D+6}$ and $p'_v = \frac{D}{3D^2-D+2}$ for $v \neq s, x$. It is easy to see that $|p_x - p'_x| = |1/(3D+1) - 1/(9D+6)|$ is already $\Omega(1/D)$.

Suppose we remove the edge between nodes $x, y \neq s$. Let $p''$ be the new PPR vector for $s$. We can verify that $p''_s = \frac{6D^3+D^2-5D}{9D^3-7D-2}, p''_x = p''_y = \frac{1}{3D+2}$, and $p''_v = \frac{3D^2-D}{9D^3-7D-2}$ for $v \neq x, y$. It is easy to see that $|p_x - p''_x| = |1/(3D+1) - 1/(3D+2)|$ is already $\Omega(1/D^2)$.

# B  Proof of Lemma 3.1

As defined in Section 2.1, let $W$ be the lazy random walk matrix of $G$.

**Lemma B.1** (Linearity of PPR vector [3]). *Given any $x \in \mathbb{R}^n$, $\mathbf{p}(x) \cdot W = \mathbf{p}(x \cdot W)$. Given any $x, y \in \mathbb{R}^n$, $\mathbf{p}(x + y) = \mathbf{p}(x) + \mathbf{p}(y)$. Given any $x \in \mathbb{R}^n, a \in \mathbb{R}$, $\mathbf{p}(a \cdot x) = a \cdot \mathbf{p}(x)$.*

Similar to Lemma 3.4 of [3], we have the following lemma.

**Lemma B.2** (Push flow operation). *Consider $n$-node graph $G = (V, E)$. Let $s \in \mathbb{R}^n$ be the starting distribution vector. Given vectors $p \in \mathbb{R}^n$ and $r \in \mathbb{R}^n$ satisfying $p = \mathbf{p}(s - r)$. For any $x \in \mathbb{R}^n$, if $p', r'$ are computed as the following:*

*1. $\forall v \in V, p'_v = p_v + \alpha \cdot x_v$,*

*2. $\forall v \in V, r'_v = r_v - x_v + (1 - \alpha)/2 \cdot \left( x_v + \sum_{(v,u) \in E} x_u/d(u) \right)$,*

*then we have $p' = \mathbf{p}(s - r')$.*

*Proof.* We can rewrite the computation of $p', r'$ as the following:

1. $p' = p + \alpha \cdot x$,

2. $r' = r - x + (1 - \alpha) \cdot x \cdot W$.

Thus, we have:

$$
\begin{aligned}
\mathbf{p}(r) &= \mathbf{p}(r - x) + \mathbf{p}(x) \\
&= \mathbf{p}(r - x) + \alpha x + (1 - \alpha)\mathbf{p}(x)W \\
&= \mathbf{p}(r - x) + \alpha x + \mathbf{p}((1 - \alpha)xW) \\
&= \mathbf{p}(r - x + (1 - \alpha)xW) + \alpha x \\
&= \mathbf{p}(r') + p' - p,
\end{aligned}
$$

where the first equality follows from linearity (Lemma B.1), the second equality follows from the definition of PPR $\mathbf{p}(x)$, the third and the forth equalities follow from linearity (Lemma B.1) again, and the last equality follows from the definition of $p'$ and $r'$.

Therefore, we have $p + \mathbf{p}(r) = p' + \mathbf{p}(r')$. Due to linearity (Lemma B.1), we have $p - \mathbf{p}(s - r) = p' - \mathbf{p}(s - r')$. Since $p = \mathbf{p}(s - r)$, $p' = \mathbf{p}(s - r')$. □

We are now able to prove Lemma 3.1.

*Proof of Lemma 3.1.* Let us first consider the running time. The algorithm has $R = O(\ln(1/\xi)/\alpha)$ iterations. In each iteration, the algorithm can visit each neighbor of each node at most once. Thus, the running time needed for each iteration is $O(|E|)$. The total running time is $O(|E| \cdot \ln(1/\xi)/\alpha)$.

Next, let us focus on the accuracy of the output. Notice that $\forall i \in [R]$, we have

1. $\forall v \in V, p_v^{(i)} = p_v^{(i-1)} + \alpha \cdot r_v^{(i-1)}$,

2. $\forall v \in V, r_v^{(i)} = (1-\alpha)/2 \cdot \left( r_v^{(i-1)} + \sum_{(v,u)\in E} r_u^{(i-1)}/d(u) \right)$.

Since $p^{(0)} = \mathbf{p}(s - r^{(0)})$, according to Lemma B.2, $\forall i \in [R], p^{(i)} = \mathbf{p}(s - r^{(i)})$. Next we want to show that each entry of $r^{(R)}$ is non-negative and is at most $\xi$. It is easy to verify that any operation during the algorithm will not create a negative value of any entry of $r^{(j)}$ for $j \in \{0, 1, \cdots, R\}$. Next consider the maximum value of $r^{(j)}$ for $j \in \{0, 1, \cdots, R\}$. The proof is by induction. It is obvious that $\|r^{(0)}\|_1 = 1$. Consider $j > 0$. We have $\|r^{(j)}\|_1 = \sum_{v \in V} r_v^{(j)} = (1-\alpha)/2 \cdot \sum_{v \in V} r_v^{(j-1)} + (1-\alpha)/2 \cdot \sum_{v \in V} \sum_{u:(u,v)\in E} r_u^{(j-1)}/d(u) = (1-\alpha)/2 \cdot \sum_{v \in V} r_v^{(j-1)} + (1-\alpha)/2 \cdot \sum_{u \in V} d(u) \cdot r_u^{(j-1)}/d(u) = (1-\alpha) \cdot \|r^{(j-1)}\|_1 \le (1-\alpha)^j$. Thus, $\|r^{(R)}\|_\infty \le \|r^{(R)}\|_1 \le (1-\alpha)^R \le \xi$.

According to Definition 2.2, since $p^{(R)} = \mathbf{p}(s - r^{(R)})$ and each entry of $r^{(R)}$ is non-negative and is at most $\xi$, $p^{(R)}$ is an $\xi$-approximate PPR vector for $\mathbf{p}(s)$. $\square$

## C  Bound of $r'_x, r'_y$ in the Proof of Theorem 3.2

**Claim C.1.** $\forall j \in [R], \|r'^{(j)}\|_1 \le (1-\alpha)^j$.

*Proof of Claim C.1.* The proof is by induction. It is obvious that $\|r'^{(0)}\|_1 = 1$. Consider $j > 0$. We have $\|r'^{(j)}\|_1 = \sum_{v \in V} r_v'^{(j)} = (1-\alpha)/2 \cdot \sum_{v \in V} r_v'^{(j-1)} + (1-\alpha)/2 \cdot \sum_{v \in V} \sum_{u:(u,v)\in E'} r_u'^{(j-1)}/d'(u) = (1-\alpha)/2 \cdot \sum_{v \in V} r_v'^{(j-1)} + (1-\alpha)/2 \cdot \sum_{u \in V} d'(u) \cdot r_u'^{(j-1)}/d'(u) = (1-\alpha) \cdot \|r'^{(j-1)}\|_1 \le (1-\alpha)^j$. $\square$

**Claim C.2.** $\forall j \in [R], \forall u \ne s, r_u'^{(j)} \le (1-\alpha)^j/D$.

*Proof of Claim C.2.* The proof is by induction. When $j = 0, \forall u \ne s, r_u'^{(0)} = 0$. Consider $j > 0$. We have $r_u'^{(j)} \le (1-\alpha)/2 \cdot r_u'^{(j-1)} + (1-\alpha)/2 \cdot \sum_{v \in V} r_v'^{(j-1)}/d'(v) \le (1-\alpha)/2 \cdot r_u'^{(j-1)} + (1-\alpha)/2 \cdot \|r'^{(j-1)}\|_1/D$. According to Claim C.1 and induction hypothesis, we have $r_u'^{(j)} \le (1-\alpha) \cdot (1-\alpha)^{j-1}/D = (1-\alpha)^j/D$. $\square$

## D  Naïve DP PPR for High Degree Graphs

---

**Algorithm 5** DPPUSHFLOW$(G, s, \alpha, \xi, \varepsilon)$

---

1: **Input:** Graph $G = (V, E)$ with minimum degree at least $D$, source node $s \in V$, teleport probability $\alpha$, precision $\xi$, DP parameter $\varepsilon$.
2: **Output:** $\varepsilon$-DP approximate PPR vector for $\mathbf{p}(s)$.
3: $p \leftarrow$ PUSHFLOW$(G, s, \alpha, \xi)$.
4: // If considering joint $\varepsilon$-DP:
5: Let $Y_1, Y_2, \cdots, Y_n$ drawn independently from Lap $\left( \frac{2(1-\alpha)}{\varepsilon \cdot \alpha \cdot D^2} \right)$
6: // Otherwise for $\varepsilon$-DP:
7: Let $Y_1, Y_2, \cdots, Y_n$ drawn independently from Lap $\left( \frac{2(1-\alpha)}{\varepsilon \cdot \alpha \cdot D} \right)$
8: Output $p + (Y_1, Y_2, \cdots, Y_n)$.

---

**Corollary D.1.** *Suppose the input graph $G$ is guaranteed to have minimum degree at least $D$. For any given source node $s$, the output of $\mathrm{DPPUSHFLOW}(G, s, \alpha, \xi, \varepsilon)$ is $\varepsilon$-DP and is a $\left(\xi, O_{\alpha,\varepsilon}(D^{-1} \ln \frac{n}{\delta})\right)$-approximate PPR for $\mathbf{p}(s)$ with probability at least $1 - \delta$ for any $\delta \in (0, 1)$. The family of (personalized) algorithms $\{\mathcal{A}_s(G) := \mathrm{DPPUSHFLOW}(G, s, \alpha, \xi, \varepsilon) \mid s \in V\}$ is joint $\varepsilon$-DP and the output of $\mathcal{A}_s(G)$ is a $\left(\xi, O_{\alpha,\varepsilon}(D^{-2} \ln \frac{n}{\delta})\right)$-approximate PPR for $\mathbf{p}(s)$ with probability at least $1 - \delta$ for any $\delta \in (0, 1)$.*

*Proof of Corollary D.1.* The $\varepsilon$-DP guarantee follows from Laplace mechanism (Theorem 2.6). The sensitivity bound is given by Theorem 3.2. Next, consider the accuracy of Algorithm 5. According to Lemma 3.1, the vector $p$ is a $\xi$-approximate PPR vector. Let $\Delta$ be the sensitivity of $p$, i.e., $\Delta = \frac{2 \cdot (1-\alpha)}{\alpha \cdot D^2}$ if joint DP is considered, and $\Delta = \frac{2 \cdot (1-\alpha)}{\alpha \cdot D}$ otherwise. Consider $i \in [n]$. By Fact 2.1, with probability at least $1 - \delta/n$, $|Y_i| \leq \frac{\Delta}{\varepsilon} \cdot \ln(n/\delta)$. By taking union bound over $i \in [n]$, with probability at least $1 - \delta$, $\max_{i \in [n]} |Y_i| \leq \frac{\Delta}{\varepsilon} \cdot \ln(n/\delta)$. Thus, the output of Algorithm 5 is a $\left(\xi, \frac{\Delta}{\varepsilon} \cdot \ln(n/\delta)\right)$-approximate PPR for $\mathbf{p}(s)$ with probability at least $1 - \delta$. $\qquad\square$

# E  Proof of Lemma 4.2

*Proof of Lemma 4.2.* According to the description of Algorithm 2, we have $\forall i \in [R], v \in V, r_v^{(i)} = \frac{1-\alpha}{2} \cdot \left( f_v^{(i)} + \sum_{u:(u,v) \in E} \frac{f_u^{(i)}}{d(u)} \right)$. Due to $h_v^{(i)} = \sum_{j=1}^{i} f_v^{(j)}$ and Observation 4.1,

$$\sum_{j=0}^{i-1} r_v^{(j)} = r_v^{(0)} + \frac{1-\alpha}{2} \cdot \left( h_v^{(i-1)} + \sum_{u:(u,v) \in E} \frac{h_u^{(i-1)}}{d(u)} \right).$$ For Observation 4.1 again, we can conclude

$$h_v^{(i)} = \min \left( r_v^{(0)} + \frac{1-\alpha}{2} \cdot \left( h_v^{(i-1)} + \sum_{u:(u,v) \in E} \frac{h_u^{(i-1)}}{d(u)} \right), d(v) \cdot T_v \right). \qquad\square$$

# F  Proof of Lemma 4.4

*Proof of Lemma 4.4.* Let us consider the running time. The algorithm has $R$ iterations. In each iteration, the algorithm can visit each neighbor of each node at most once. Thus, the running time for each iteration is $O(|E|)$. The total running time is $O(|E| \cdot R) = O(|E| \cdot \ln(1/\xi)/\alpha)$.

Let $D = \max \left( 1/(\alpha T_s), \sqrt{1/(\alpha T_u)} \right)$ $(u \neq s)$. In the remaining of the proof, we consider the input graph $G$ with minimum degree at least $D$. Firstly, we observe that the value $R$ is the same in Algorithm 1 and Algorithm 2, i.e., both algorithms have the same number of iterations. Then notice that if $f_v^{(i)}$ is equal to $r_v^{(i-1)}$ for all $i$ and $v$, then we have $\forall i \in [R]$,

1. $\forall v \in V, p_v^{(i)} = p_v^{(i-1)} + \alpha \cdot r_v^{(i-1)}$,

2. $\forall v \in V, r_v^{(i)} = (1-\alpha)/2 \cdot \left( r_v^{(i-1)} + \sum_{(v,u) \in E} r_u^{(i-1)}/d(u) \right)$.

Therefore, when $f_v^{(i)}$ is equal to $r_v^{(i-1)}$ for all $i$ and $v$, vectors $p^{(i)}, r^{(i)}$ in both Algorithm 1 and Algorithm 2 are equal respectively. Thus, when $f_v^{(i)}$ is equal to $r_v^{(i-1)}$ for all $i$ and $v$, the output of Algorithm 2 is the same as the output of Algorithm 1. It suffices to show that $\forall i \in [R], v \in V, f_v^{(i)} = r_v^{(i-1)}$ when $G$ has minimum degree at least $D$.

The proof is by contradiction. Consider the first time that $f_v^{(i)} \neq r_v^{(i-1)}$ during the execution of Algorithm 2. We have $d(v) \cdot T_v - h_v^{(i-1)} < r_v^{(i-1)}$. Since it is the first time $f_v^{(i)} \neq r_v^{(i-1)}$, we know $h_v^{(i-1)} = \sum_{j=0}^{i-2} r_v^{(j)}$ according to Observation 4.1. Therefore, we know that $\sum_{j=0}^{i-1} r_v^{(j)} > d(v) \cdot T_v$. If $v \neq s$, according to the choice of $T_v$, we have $\sum_{j=0}^{i-1} r_v^{(j)} > D/(\alpha \cdot D^2) = 1/(\alpha D)$ which contradicts to Claim C.2. If $v = s$, according to the choice of $T_v$, we have $\sum_{j=0}^{i-1} r_v^{(j)} > D/(\alpha \cdot D) = 1/\alpha$

which contradicts to Claim C.1. Thus, we always have $\forall i \in [R], v \in V, f_v^{(i)} = r_v^{(i-1)}$ when $G$ has minimum degree at least $D$. We conclude that the output of Algorithm 2 is the same as the output of Algorithm 1 in this case. According to Lemma 3.1, the output is a $\xi$-approximate PPR vector. $\square$

## G Proof of Corollary 4.6

*Proof of Corollary 4.6.* Since the joint sensitivity and non-joint sensitivity of Algorithm 2 are given by Theorem 4.3. Thus, the $\varepsilon$-DP and joint $\varepsilon$-DP guarantees follow from the Laplace mechanism (Theorem 2.6).

Next, consider the accuracy of Algorithm 3. If considering joint $\varepsilon$-DP, according to Lemma 4.4, the output $p$ of PUSHFLOWCAP (Algorithm 2) is a $\xi$-approximate PPR vector when the minimum degree of $G$ is at least $\sqrt{(2 \cdot (2 - \alpha))/(\alpha \cdot \sigma)}$. Otherwise, the output $p$ of PUSHFLOWCAP (Algorithm 2) is a $\xi$-approximate PPR vector when the minimum degree of $G$ is at least $(2 \cdot (2 - \alpha))/(\alpha \cdot \sigma)$. Consider $i \in [n]$. By Fact 2.1, with probability at least $1 - \delta/n$, $|Y_i| \leq \frac{\sigma}{\varepsilon} \cdot \ln(n/\delta)$. By taking union bound over $i \in [n]$, with probability at least $1 - \delta$, $\max_{i \in [n]} |Y_i| \leq \frac{\sigma}{\varepsilon} \cdot \ln(n/\delta)$. Thus, the output of Algorithm 3 is a $\left(\xi, \frac{\sigma}{\varepsilon} \cdot \ln(n/\delta)\right)$-approximate PPR for $\mathbf{p}(s)$ with probability at least $1 - \delta$. $\square$

## H Proof of Theorem 5.1

*Proof of Theorem 5.1.* For $v \in V$, consider $|\max(\log(p_v \cdot n), 0) - \max(\log(p'_v \cdot n), 0)|$. There are several cases. Without loss of generality, we suppose $p_v > p'_v$. In the first case, if both $p_v, p'_v \leq 1/n$, then we have $|\max(\log(p_v \cdot n), 0) - \max(\log(p'_v \cdot n), 0)| = 0 \leq \log(1 + |p_v - p'_v| \cdot n)$. In the second case, $p_v > 1/n$ and $p'_v \leq 1/n$. Then, $|\max(\log(p_v \cdot n), 0) - \max(\log(p'_v \cdot n), 0)| = \log(1 + (p_v - 1/n) \cdot n) \leq \log(1 + |p_v - p'_v| \cdot n)$. In the third case, both $p_v, p'_v > 1/n$. Then $|\max(\log(p_v \cdot n), 0) - \max(\log(p'_v \cdot n), 0)| = \log(p_v/p'_v) = \log(1 + (p_v - p'_v)/p'_v) \leq \log(1 + |p_v - p'_v| \cdot n)$. By combining the above cases, we always have:

$$\begin{aligned} &|\max(\log(p_v \cdot n), 0) - \max(\log(p'_v \cdot n), 0)| \\ &\leq \log(1 + |p_v - p'_v| \cdot n) \end{aligned} \tag{3}$$

Now, we are able to bound $\|w - w'\|_1$.

$$\begin{aligned} &\|w - w'\|_1 \\ &= \sum_{i=1}^{k} |\sum_{v \in V: h_k(v) = i} h_{sgn}(v) \cdot (\max(\log(p_v \cdot n), 0) \\ &\quad - \max(\log(p'_v \cdot n), 0))| \\ &\leq \sum_{v \in V: p_v \neq p'_v} |\max(\log(p_v \cdot n), 0) - \max(\log(p'_v \cdot n), 0)| \\ &\leq \sum_{v \in V: p_v \neq p'_v} \log(1 + |p_v - p'_v| \cdot n) \\ &\leq m \cdot \log(1 + \|p - p'\|_1 \cdot n/m), \end{aligned}$$

where the first inequality follows from triangle inequality, the second inequality follows from Equation (3), and the last inequality follows from the concavity of $\log(\cdot)$. $\square$

## I Differentially Private InstantEmbedding via Sparse Personalized PageRank

**Theoretically Improved InstantEmbedding via Sparse Personalized PageRank.** Notice that due to Theorem 5.1, a sparser approximate PPR may give lower sensitivity of the InstantEmbedding. Therefore, we show how the embedding algorithm can be further improved by sparsifying the PPR vector in a differentially private way. The sparsification procedure is reported in Algorithm 6 which keeps large entries with good probabilites and will drop small entries of the input vector.

**Lemma I.1.** DPSPARSIFICATION$(p, \sigma, \varepsilon, \gamma)$ *is $\varepsilon$-DP.*

*Proof.* Consider a neighboring vector $p'$ of $p$ i.e., $\|p - p'\|_1 \le \sigma$. Let $S$ and $S'$ be the output of DPSPARSIFICATION$(p, \sigma, \varepsilon, \gamma)$ and DPSPARSIFICATION$(p', \sigma, \varepsilon, \gamma)$ respectively. For $i \in [n]$, $\Pr[i \in S]/\Pr[i \in S'] = \int_{-\infty}^{p_i} \frac{\varepsilon}{2\sigma} e^{-\frac{\varepsilon}{\sigma}|x-\gamma|} \mathrm{d}x / \int_{-\infty}^{p'_i} \frac{\varepsilon}{2\sigma} e^{-\frac{\varepsilon}{\sigma}|x-\gamma|} \mathrm{d}x$. Notice that $\exp\left(-\frac{\varepsilon}{\sigma}|x-\gamma|\right) / \exp\left(-\frac{\varepsilon}{\sigma}|x+p'_i - p_i - \gamma|\right) \le \exp\left(\frac{\varepsilon}{\sigma}|p'_i - p_i|\right)$. Therefore, $\Pr[i \in S]/\Pr[i \in S'] \le \exp\left(\frac{\varepsilon}{\sigma}|p'_i - p_i|\right)$. By similar argument, we can also prove that $\Pr[i \notin S]/\Pr[i \notin S'] \le \exp\left(\frac{\varepsilon}{\sigma}|p'_i - p_i|\right)$. Hence, $\forall X \subseteq [n]$, $\Pr[S = X]/\Pr[S' = X] \le \exp\left(\frac{\varepsilon}{\sigma}\|p' - p\|_1\right) \le \exp(\varepsilon)$ that concludes the proof. $\square$

**Lemma I.2** (Sparisity of DPSPARSIFICATION$(p, \sigma, \varepsilon, \gamma)$)**.** *If $\|p\|_1 \le 1$ and $\gamma \ge \frac{3\sigma}{\varepsilon}\ln(n)$, with probability at least $1 - 1/n$, the output $S$ of DPSPARSIFICATION$(p, \sigma, \varepsilon, \gamma)$ satisfies (1) $|S| \le 3/\gamma$, (2) $\forall i \in S, p_i \ge \gamma/3$, (3) $\forall i$ with $p_i \ge 2\gamma, i \in S$.*

*Proof.* If $p_i < \gamma/3$, since $\gamma \ge \frac{3\sigma}{\varepsilon}\ln(n)$, the probability that $i$ is added to $S$ is at most $\frac{1}{2}\exp(-2\ln n) = 1/(2n^2)$. By taking union bound over all $i \in [n]$, with probability at least $1/(2n)$, $\forall i$ with $p_i < \gamma/3$, $i \notin S$. Since $\|p\|_1 \le 1$, $|S| \le 3/\gamma$. If $p_i \ge 2\gamma$, the probability that $i$ is added to $S$ is at least $1 - 1/(2n^2)$. By taking union bound over all $i \in [n]$, with probability at least $1/(2n)$, $\forall i$ with $p_i \ge 2\gamma, i \in S$. $\square$

As a side result, we obtain DP sparse approximate PPR vector by applying composition (Theorem 2.7) of the sparsification (Algorithm 6) and Laplace mechanism (Theorem 2.6) on our senstivity-bounded PPR vector (Algorithm 2). We refer readers to Appendix J for more details. In Algorithm 7, we show

---

**Algorithm 6** DPSPARSIFICATION$(p, \sigma, \varepsilon, \gamma)$

1: **Input:** An (approximate PPR) vector $p \in \mathbb{R}^n$, a sensitivity upper bound $\sigma$ of $p$, a parameter $\varepsilon$ for DP, and a threshold $\gamma$.
2: **Output:** An $\varepsilon$-differentially private set of indices $S \subseteq [n]$.
3: Initialize $S \leftarrow \emptyset$.
4: For each $i \in [n]$, if $p_i \le \gamma$, add $i$ into $S$ with probability $\frac{1}{2} \cdot \exp\left(-\frac{\varepsilon}{\sigma} \cdot (\gamma - p_i)\right)$, otherwise add $i$ into $S$ with probability $1 - \frac{1}{2} \cdot \exp\left(\frac{\varepsilon}{\sigma} \cdot (\gamma - p_i)\right)$.
5: Output $S$.

---

how to get DP InstantEmbedding via DP sparse approximate PPR vector. We record the theoretical

---

**Algorithm 7** DPEMBEDDINGSPARSE$(G, s, \alpha, \xi, \sigma, k, \varepsilon, type)$

1: **Input:** Graph $G = (V, E)$, source $s \in V$, teleport probability $\alpha$, precision $\xi$, sensitivity parameter $\sigma$, embedding dimension $k$, DP parameter $\varepsilon$, and $type \in \{joint, non\text{-}joint\}$ indicating whether joint $\varepsilon$-DP or $\varepsilon$-DP is required.
2: **Output:** $\varepsilon$-DP $k$-dimensional embedding vector.
3: $\varepsilon_0 \leftarrow \varepsilon/2$.
4: $\hat{p} \leftarrow$ PUSHFLOWCAP$(G, s, \alpha, \xi, \sigma, type)$.
5: $S \leftarrow$ DPSPARSIFICATION $\left(\hat{p}, \sigma, \varepsilon_0, \frac{3\sigma}{\varepsilon_0}\ln n\right)$.
6: Construct $p$ such that $p_i = \hat{p}_i$ for $i \in S$ and $p_i = 0$ for $i \in [n] \setminus S$.
7: $w \leftarrow$ INSTANTEMBEDDING$(p, k)$.
8: Draw $Y_1, Y_2, \cdots, Y_k$ independently from Lap$(|S| \cdot \log(1 + \sigma \cdot n/|S|)/\varepsilon_0)$.
9: Output $w + (Y_1, Y_2, \cdots, Y_k)$.

---

guarantees of the algorithm in Theorem I.3.

**Theorem I.3.** *The family of (personalized) algorithms $\{\mathcal{A}_s(G) :=$ DPEMBEDDINGSPARSE$(G, s, \alpha, \xi, \sigma, k, \varepsilon, joint) \mid s \in V\}$ is joint $\varepsilon$-DP, and DPEMBEDDINGSPARSE$(G, s, \alpha, \xi, \sigma, k, \varepsilon, non\text{-}joint)$ is $\varepsilon$-DP with respect to $G$ for any $s \in V$. In addition, if the input graph $G$ has a minimum degree at least $D$, then the joint $\varepsilon$-DP (resp. $\varepsilon$-DP) output is a $(\xi, O_{\alpha,\varepsilon}(\sigma\log(n)))$-approximate PPR for $\mathbf{p}(s)$ with $O_{\alpha,\varepsilon}\left(\frac{1}{\sigma\log n}\right)$ non-zero entries with probability at least $1 - O(1/n)$ when $\sigma \ge \Omega_\alpha(1/D^2)$ (resp. $\sigma \ge \Omega_\alpha(1/D)$).*

*Proof.* Firstly, let us consider the DP guarantee. According to Lemma I.1, $S$ is $\varepsilon_0$-DP. Since $p$ has $|S|$ non-zero entries, the sensitivity of $w$ is at most $|S| \cdot \log(1 + \sigma \cdot n/|S|)$. Due to Laplace mechanism

(Theorem 2.6), given $S$, the final output is $\varepsilon_0$-DP. Since $S$ is also $\varepsilon_0$-DP, according to composition theorem (Theorem 2.7), the overall algorithm is $(\varepsilon_0 + \varepsilon_0)$-DP, i.e., $\varepsilon$-DP.

Due to the proof of Theorem J.1 (see Appendix J), with probability at least $1 - O(1/n)$, $p$ is a $(\xi, O(\sigma \log(n)/\varepsilon))$-approximate PPR vector for $\mathbf{p}(s)$, and $|S| \leq O\left(\frac{\varepsilon}{\sigma \log n}\right)$. Due to Fact 2.1 and union bound, with probability at least $1 - O(1/n)$, $\max_{i \in [k]} |Y_i| \leq |S| \cdot \log\left(1 + \sigma \cdot n/|S|\right)/\varepsilon_0 \cdot \log n \leq O(\sigma^{-1}\log(1 + \sigma \cdot n))$. $\qquad\square$

## J Differetially Private Sparse Approximate PPR

**Theorem J.1.** *Given source $s$, teleport probability $\alpha$, precision $\xi$, sensitivity bound $\sigma$ and DP parameter $\varepsilon$, there is an algorithm which is always $\varepsilon$-DP with respect to the input $n$-node graph $G$ and in addition outputs $(\xi, O_{\alpha,\varepsilon}(\sigma \ln n))$-approximate PPR vector with $O_{\alpha,\varepsilon}(1/(\sigma \ln n))$ non-zero entries for $\mathbf{p}(s)$ with probability at least $1 - O(1/n)$ when $G$ has minimum degree at least $\Omega_\alpha(1/\sigma)$. There is a family of (personalized) algorithms $\{\mathcal{A}_1, \mathcal{A}_2, \cdots, \mathcal{A}_n\}$ which is joint $\varepsilon$-DP with respect to the input $n$-node graph $G$ and in addition $\forall s \in V$, $\mathcal{A}_s(G)$ outputs $(\xi, O_{\alpha,\varepsilon}(\sigma \ln n))$-approximate PPR vector with $O_{\alpha,\varepsilon}(1/(\sigma \ln n))$ non-zero entries for $\mathbf{p}(s)$ when $G$ has minimum degree at least $\Omega_\alpha(\sqrt{1/\sigma})$ with probability at least $1 - O(1/n)$.*

The algorithm that outputs the $\varepsilon$-DP sparse approximate PPR is given in Algorithm 8.

---

**Algorithm 8** DPSPARSEPPR($G, s, \alpha, \xi, \sigma, \varepsilon, joint/non\text{-}joint$)

---

1: **Input:** Graph $G = (V, E)$, source $s \in V$, teleport probability $\alpha$, precision $\xi$, sensitivity bound $\sigma$, DP parameter $\varepsilon$, and $type \in \{joint, non\text{-}joint\}$ indicating whether joint $\varepsilon$-DP or $\varepsilon$-DP is considered.
2: **Output:** $\varepsilon$-DP or joint $\varepsilon$-DP approximate PPR vector for $\mathbf{p}(s)$.
3: $\varepsilon_0 \leftarrow \varepsilon/2$.
4: $\widehat{p} \leftarrow$ PUSHFLOWCAP($G, s, \alpha, \xi, \sigma, type$).
5: $S \leftarrow$ DPSPARSIFICATION $\left(\widehat{p}, \sigma, \varepsilon_0, \frac{3\sigma}{\varepsilon_0} \ln n\right)$.
6: For each $i \in S$, let $Y_i$ be drawn independently from Lap $(\sigma/\varepsilon_0)$
7: Construct $p$ such that $p_i = \widehat{p}_i$ for $i \in S$. Let $p_i = Y_i = 0$ for $i \in [n] \setminus S$.
8: Output $p + (Y_0, Y_1, \cdots, Y_n)$.

---

*Proof of Theorem J.1.* According to Theorem 4.3, the (joint) sensitivity of $\widehat{p}$ is $\sigma$. According to Lemma I.1, set $S$ is $\varepsilon_0$-DP. For any fixed set $S$, the sensitivity of $p$ is always bounded by the sensitivity of $\widehat{p}$. Therefore, given $S$, $p + (Y_0, Y_1, \cdots, Y_n)$ is $\varepsilon_0$-DP due to Laplace mechanism (Theorem 2.6). According to the composition theorem, Theorem 2.7, the final output is $(\varepsilon_0 + \varepsilon_0)$-DP which is $\varepsilon$-DP. According to Lemma I.2, with probability at least $1 - 1/n$, the number of non-zero entries of the output is $O(\varepsilon/(\sigma \ln n))$.

Next, let us consider the accuracy of the output. Suppose joint $\varepsilon$-DP is considered. If $G$ has minimum degree at least $\Omega_\alpha(\sqrt{1/\sigma})$, according to Lemma 4.4, $\widehat{p}$ is a $\xi$-approximate PPR vector for $\mathbf{p}(s)$. Suppose $\varepsilon$-DP is considered. If $G$ has minimum degree at least $\Omega_\alpha(1/\sigma)$, according to Lemma 4.4, $\widehat{p}$ is a $\xi$-approximate PPR vector for $\mathbf{p}(s)$.

Notice that $\gamma = \frac{3\sigma}{\varepsilon_0} \cdot \ln n$. According to Lemma I.2, $\|p - \widehat{p}\|_\infty \leq O(\gamma)$. According to Fact 2.1, with probability at least $1 - 1/n$, $\max_{i \in [n]} |Y_i| \leq O(\gamma)$. Thus, with probability at least $1 - O(1/n)$, the output is a $(\xi, O(\gamma))$-approximate PPR vector for $\mathbf{p}(s)$. $\qquad\square$

## K Additional Experimental Results

**Additional baselines** We also considered DPNE [30] in our evaluation as a potential baseline. However, the DPNE algorithm as described [30] is not DP in the setting of edge-DP or joint-DP unless certain assumptions on the input graph are made.[6] Since the paper does not provide edge-DP or joint-edge DP on *arbitrary* inputs, like our paper, we omit it from our empirical evaluation.

---

[6]This has been confirmed in a personal communication with the authors.

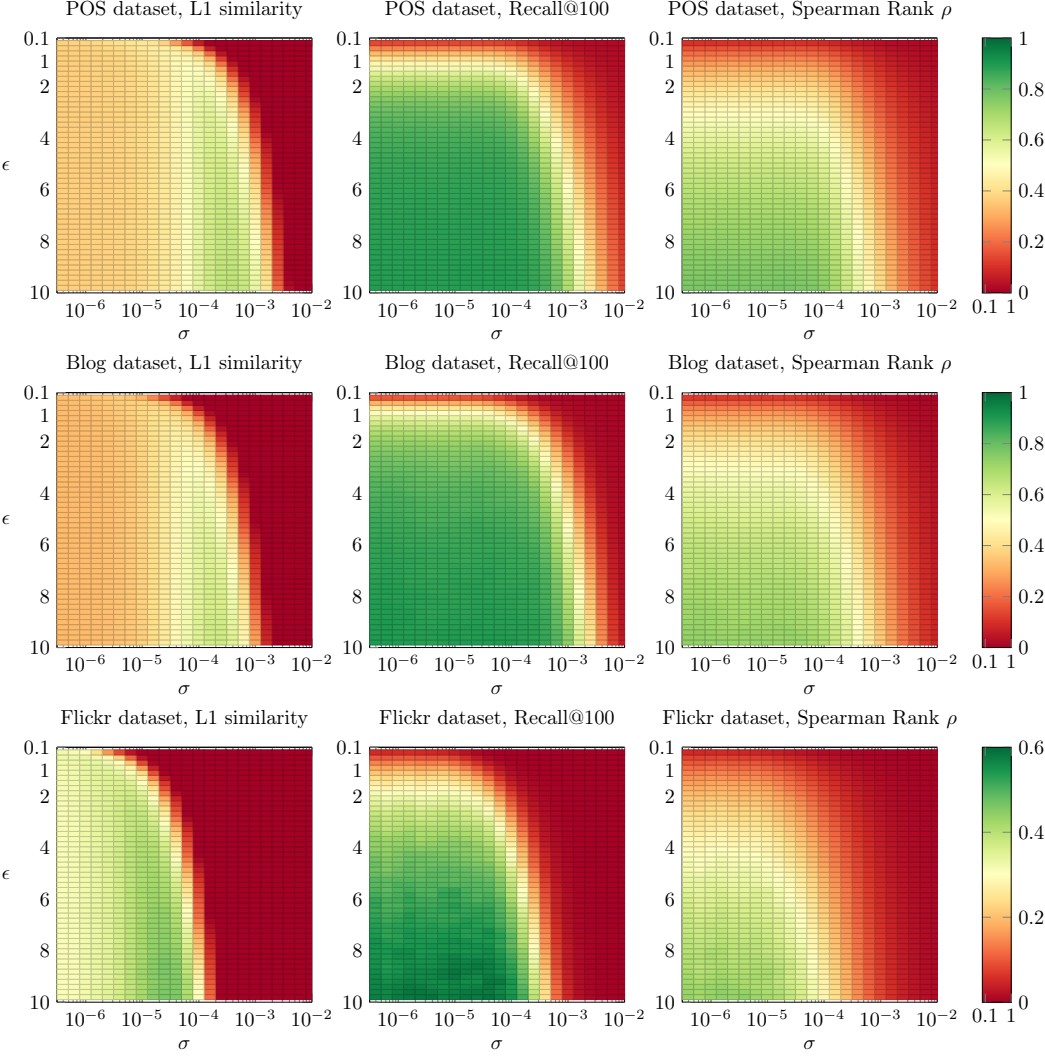

Figure 3: Sensitivity to the $\sigma$ parameter measured across three different datasets and three different metrics – $L_1$ similarity, Recall@100, and Spearman rank correlation coefficient $\rho$. Best viewed in colour.

**Details on the datasets** We provide more detailed description of the datasets. PPI is a protein-protein interaction dataset, where labels represent hallmark gene sets of specific biological states. Blogcatalog is a social networks of bloggers, where labels are self-identified topics of their blogs. Flickr is a photo-based social network, where labels represent self-identified interests of users and edges represent messages between users.

**Additional studies on parameter settings.** We report in Figure 3 an extended version of our analysis of the accuracy of DP PPR in approximating the PPR rankings. We are especially interested in a reliable mechanism for setting the parameter $\sigma$ of our algorithm. We present the result according to three different metrics: $L_1$ similarity $(1 - L_1$ distance$)$, Recall@100, and Spearman rank correlation coefficient $\rho$. The most indicative metric of the three is Recall@100, since it evaluates the quality of the nearest neighborhood of the node. We observe good performance across wide range of $\sigma$.

We report in Figure 4 comparative analysis of the total residual in the joint DP and non-joint DP versions. We observe that for almost all range of the sensitivity parameter $\sigma$, joint DP offers more push compared to non-joint DP version of the algorithm.

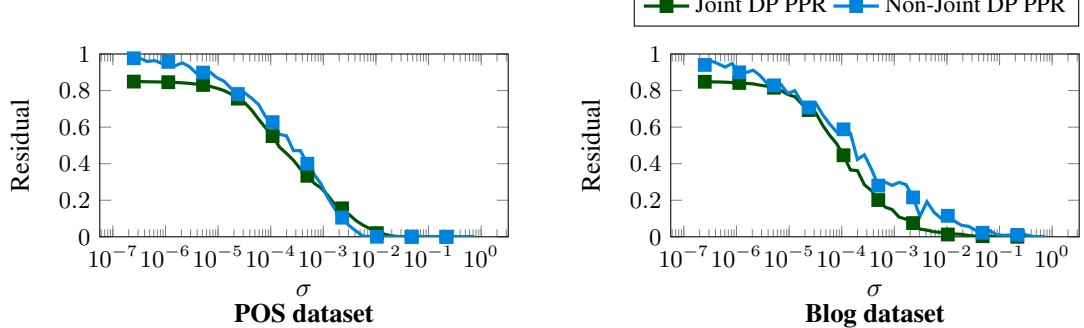

Figure 4: PPR residual statistics on two datasets.