# OpenReview forum: "Differentially Private Graph Learning via Sensitivity-Bounded Personalized PageRank"
_NeurIPS.cc/2022/Conference — NeurIPS 2022 Accept_

### Official Review · Reviewer_gvyQ · 2022-07-10

**Rating:** 7
**Confidence:** 2
**Soundness:** 4 excellent
**Presentation:** 2 fair
**Contribution:** 3 good

**Summary:**

This paper focuses on computing approximate Personalized PageRank (PPR) vectors with differential privacy (DP) in graph representation learning. First, the authors analyze the non-private version $\xi$-approximate PPR push algorithm with sensitivity. Then, the authors develop a variant of the push algorithm with bounded sensitivity called PushFlowCap. By applying Laplace mechanism with PushFlowCap, they also propose an algorithm called DPPushFlowCap to obtain an non-joint or joint $\epsilon$-DP approximate PPR vector. Finally, based on InstantEmbedding, the authors provide a sensitivity-bounded version for graph embedding.

**Questions:**

1. Why do the authors specifically choose the Laplace distribution? There are no citations or explanations in Section 2 from line 78 to line 88.

**Limitations:**

The authors have sufficiently addressed limitations of this work. In Section 7, the authors concern the safety of any application of this work.

**Strengths And Weaknesses:**

Strengths:

- The theoretical foundation is solid.

- The algorithms are written clearly.

Weaknesses:

- The idea of this paper is based on an assumption that a GNN with the Personalized PageRank filter is highly sensitive to edge changes on a graph such as a single edge addition or removal. However, according to research works about stability for polynomial spectral graph filters[1,2] which include the Generalized PageRank filter[3,4], a generalization of all PageRank filter variants, polynomial spectral graph filters are robust for edge modifications so long as the endpoint degrees are high. Therefore, the assumption for this work is not entirely valid.

- The introduction of relations among the $(\xi,\eta)$-approximate PPR vector, non-joint and joint DPPPR vector, PushFlowCap algorithm, and DPPushFlowCap algorithm are vague.

References:
1. Kenlay, H., Thanou, D., & Dong, X. (2020). On The Stability of Polynomial Spectral Graph Filters. In ICASSP 2020 - 2020 IEEE International Conference on Acoustics, Speech and Signal Processing (ICASSP) (pp. 5350–5354).
2. Kenlay, H., Thanou, D., & Dong, X. (2021). Interpretable Stability Bounds for Spectral Graph Filters. In Proceedings of the 38th International Conference on Machine Learning (pp. 5388–5397). PMLR.
3. Baeza-Yates, R., Boldi, P., & Castillo, C. (2006). Generalizing PageRank: Damping Functions for Link-Based Ranking Algorithms. In Proceedings of the 29th Annual International ACM SIGIR Conference on Research and Development in Information Retrieval (pp. 308–315). Association for Computing Machinery.
4. Gleich, D. (2015). PageRank Beyond the Web. SIAM Review, 57(3), 321–363.

---

> ### Author Response · Authors · 2022-08-02
> **Response to Reviewer gvyQ**
>
> Thanks for the review! Below we address your comments.
>
> -"Therefore, the assumption for this work is not entirely valid": Our goal is to develop a differentially private personalized pagerank algorithm such that the DP guarantee does not depend on the input graph. In other words, our algorithm works for the worst case guarantee and does not have any assumption on the input graph. This is standard and considered necessary in the privacy world as we want the users to know that they will be protected no matter what their data looks like.
>
> Notice that we need to be able to make theoretical guarantees of worst-case stability in order to provide privacy protection in  the differential privacy framework. As we show in our paper, the standard PPR is highly sensitive to a single edge addition in the worst case and we design our PPR approximation algorithm to provide provable bounds on sensitivity.
> Moreover, in Appendix A, we show that the sensitivity of the ground truth PPR can be large which implies that the theoretical guarantee of our algorithm is almost tight.
>
> -"The introduction of relations among the (\xi,\eta)-approximate PPR vector, non-joint and joint DPPPR vector, PushFlowCap algorithm, and DPPushFlowCap algorithm are vague.": We presented formal definitions of all notions we used. In high level, (\xi,\eta)-approximation is used for describe the accuracy guarantee (this is a standard notion in the literature of approximations of PPR). Joint DP is a relaxation of DP such that the algorithm does not need to protect the information of edges incident to the source user as they are user data that can be used for personalizing the results of the user themselves. PushFlowCap algorithm is our novel contribution which presented a PPR algorithm with target bounded sensitivity. DPPushFlowCap is the private algorithm obtained by applying Laplace mechanism on the low sensitivity PushFlowCap algorithm.
>
> -"Why do the authors specifically choose the Laplace distribution": Laplace mechanism is a standard differential privacy technique which converts a low sensitivity algorithm to a DP algorithm. In DP it is impossible to use any noise distribution but instead it is needed to choose the correct noise distribution depending on the sensitivity of the input. We refer to standard texts on the reason for choosing Laplace (see for instance “Algorithmic foundations of differential privacy”).

---

### Official Review · Reviewer_61St · 2022-07-11

**Rating:** 6
**Confidence:** 5
**Soundness:** 3 good
**Presentation:** 2 fair
**Contribution:** 3 good

**Summary:**

In this paper, the authors propose an algorithm which outputs an approximate PPR and
has provably bounded sensitivity to input edges.

**Questions:**

Why is their no theoretical proofs in the paper?

Why is comparative analysis lacking from the Experiments?

**Strengths And Weaknesses:**

Pros:

Paper is organized well

Paper topic is interesting

Cons:

Algorithms theoretical bounds are not studied (NP-Complete etc)

Experimental analysis is brief and does not give comparative analysis to SOA algorithms

---

> ### Author Response · Authors · 2022-08-02
> **Response to Reviewer 61St**
>
> Thanks for the review! Below we address your comments.
>
> -"Algorithms theoretical bounds are not studied (NP-Complete etc)", "Why is there no theoretical proof in the paper":
> We would like to point out that our paper has a comprehensive theoretical analysis and many proofs. Most of the proofs have been moved to the supplementary material for lack of space but our algorithm has clear theoretical statements in the main paper. In particular, we show the running time of our algorithms in our statements (see e.g., Lemma 4.4) and we also show all the theoretical guarantees for both DP (Corollary 4.6) and accuracy (e.g., Lemma 4.4). We have detailed and complete proofs for all of our statements in the paper (they can be found in the appendix in the supplementary) and we mentioned this in the main body of the paper. Moreover, we observe that in general computing PPR is not NP-Complete, and that we always provide polynomial time (actually near linear time) algorithms.
>
> -”Experimental analysis is brief and does not give comparative analysis to SOA algorithms”: To the best of our knowledge, we are the first to study the DP PPR problem. Thus there is no previous work explicitly studying the problem. We compared our algorithm with the most natural randomized response baseline algorithm and we believe this baseline was the best previous algorithm for the DP PPR problem which we studied. Please check the reply to the reviewer SocG for more details on the choice of the baselines.

---

### Official Review · Reviewer_SocG · 2022-07-12

**Rating:** 6
**Confidence:** 4
**Soundness:** 4 excellent
**Presentation:** 3 good
**Contribution:** 3 good

**Summary:**

This paper studies the Personalized PageRank (PPR) problem (which is a very popular quantity
used in graphmining) under differential privacy constraints. The first contribution of
the paper involves analyzing the sensitivity of a prior algorithm for PPR, Push-Flow,
in terms of the min-degree of the graph; this could be large if the min-degree is low.
The authors design a natural variant, PUSHFLOWCAP, which has a better sensitivity,
allowing the Laplace mechanism to give privacy bounds. Finally, the authors combine their
method with a recent algorithm for embeddings. The methods are evaluated on real-world datasets.

**Questions:**

Is the notion of (\xi, \eta)-approximate PPR well established? What is the impact of the \eta
parameter on the utility of PRR in practice?

There are a lot of symbols, some of which are defined as they are encountered or after. For instance,
T_v is not yet defined in Obs 4.1 and Lemma 4.2, making it hard to parse. A table of notation
might make the paper more readable.  The statements with join/non-joint, while concise,
are also harder to read.

Section 6: while the values of hyperparameters \xi, \alpha and k might make sense in a non-private
setting, it is not clear if those are the best choices for this paper. It would be interesting
to see how they are impacted under privacy, and it might provide a way to improve the results
further.

Section 6.1: it would be useful to define the metrics.

Section 6.2: for evaluating embeddings, it might be useful to compare with the DPNE method of
(Xu et al., PAKDD 2018), which gives an algorithm for this problem. Also, how is micro-F1 defined?
The results in Figure 2 are not particularly great. Maybe considering larger \epsilon is helpful

**Limitations:**

The discussion is limited but seems ok

**Strengths And Weaknesses:**

PPR is a very commonly used metric, and differentially private algorithms are not known. So the
paper is studying an important problem. The main technical idea is quite nice, and pretty natural.
The sensitivity analysis of the Push-Flow and PUSHFLOWCAP algorithms is technically
interesting, though not particularly involved.

The statements in the final results are somewhat complicated since the utility analysis
holds under some conditions. These would become stronger if a lower bound could be shown.

The experiments section could be strengthened. While there is no private algorithm for PPR,
randomized response seems like too weak a baseline.

The presentation could be improved quite a bit. The authors could simplify the notation and present
it in a way that the symbols are easier to parse.

---

> ### Author Response · Authors · 2022-08-02
> **Response to Reviewer SocG**
>
> Thanks for the review! Below we address your questions.
>
> -"These would become stronger if a lower bound could be shown": We indeed have a lower bound (see Appendix A in the supplementary material). Our sensitivity analysis is actually tight. According to Corollary 4.6, our approximation guarantee holds when the sensitivity parameter >= 1/D^2 for joint edge DP and >= 1/D for edge DP where D is the minimum degree of the graph. This matches the lower bound (we also mentioned this in section 1.1) shown in Appendix A (see the supplementary material). In particular we show that the ground truth PPR has sensitivity Omega(1/D) and joint sensitivity Omega(1/D^2). Our results are actually theoretically optimal up to constant factors. The implication of Corollary 4.6 is hence tight, and it is impossible to have a good theoretical approximation guarantee if sigma < o(1/D) for edge DP or sigma < o(1/D^2) for joint edge level DP.
>
> -"randomized response seems like too weak a baseline": To the best of our knowledge, we are the first to study DP PPR problem. We considered all reasonable baselines we could come up with and reported the ones that performed best against us. In fact we also tried computing the standard PPR vector and adding Laplace noise using the optimal sensitivity bound of the PPR vector (which is ~1). The performance of this baseline was much poorer than the randomized response baseline and close to random so we omitted it. We will mention it in the paper.
>
> -”Is the notion of (\xi, \eta)-approximate PPR well established?”: Yes. Note that \xi denotes the error introduced by the residual and \eta denotes the error introduced by the PPR vector itself. Two types of errors are well studied in the literature. For residual error \xi, see e.g., [3] Andersen et. al. 2007. For PPR vector error \eta, see e.g., [16] Hou et. al. 2021.
>
> -”hyperparameters \xi, \alpha and k”: These parameters are from the non-private algorithms already and well accepted in the literature of PPR. In this work we show how to provide a private algorithm for any choice of \xi, \alpha and k etc. Studying the impact of these hyper parameters on utility is interesting but is out of the scope of this work (as there is an extensive literature on PPR). -”it would be useful to define the metrics”:
> We use Recall@100 and NDCG to compare DP PPR to non-private PPR computation. Recall@100 measures the number of nodes that were correctly raked in the top 100. NDCG, on the other hand, considers the rankings of all nodes, and aggregates them into a single metric that prioritizes most relevant nodes.
>
> For measuring node classification accuracy, we use the micro-averaged F1 score to be consistent with the node embedding literature [25,28]. See below for the definition of that metric.
>
> https://en.wikipedia.org/wiki/Precision_and_recall
> https://en.wikipedia.org/wiki/Discounted_cumulative_gain#Normalized_DCG
>
> For a textbook reference, see “Recommender Systems: The Textbook" by Charu Aggarwal, pp. 245
>
> -”it might be useful to compare with the DPNE method”:  Thanks for the suggestion. We considered DPNE in our evaluation as it can (at a first sight) seem a valid comparison to our algorithm. However, the DPNE algorithm as described in the paper is not DP in the setting of edge-dp or joint-dp unless certain (non explicit) assumptions on the input are made.
>
> We contacted the authors of the DPNE paper and confirmed that their paper does not provide DP as in edge-dp or joint-edge dp on arbitrary inputs like our paper. Following their guidance, we attempted to modify their algorithm to implement a version of DPNE that is at joint-DP and thus comparable to our joint-DP algorithm. Unfortunately we were not able to obtain any performance not significantly worse than than the randomized response baseline. Given that our work (which is a valid DP algorithm without further assumptions) can’t be fairly compared to the algorithm in the paper nor their paper is easily adaptable to our settings we decided to not report such results. Given the interest of the reviewers, we will clarify this in the final version of our paper by adding a section about DPNE.
>
> -”how is micro-F1 defined”:
> Micro-F1 (micro-averaged F1 score) is a standard measure used in the literature. It extends the F1-score to multiclass classification by averaging results biased by class frequency -- so all classes are treated as equally important. In the charts presented in the paper, we multiply the values by 100 for reader’s convenience. Please see as a reference: https://en.wikipedia.org/wiki/F-score#Extension_to_multi-class_classification.

---

### Official Review · Reviewer_MDNv · 2022-07-13

**Rating:** 3
**Confidence:** 4
**Soundness:** 3 good
**Presentation:** 3 good
**Contribution:** 2 fair

**Summary:**

The paper proposes algorithms that compute personalized PageRank (PPR) under edge differential privacy (DP), and extends the algorithms to compute graph embeddings.

**Questions:**

N/A

**Strengths And Weaknesses:**

The paper studies an interesting problem, and is relatively well written. However, there are several major issues with the paper.

First, the proposed PPR algorithm is relatively straightforward: it computes the PPR values and then adds noise based on the sensitivity of the values. The sensitivity analysis is mostly based on the triangle inequality and is rather loose. As a consequence, the significance of the algorithm is unclear.

Second, the proposed algorithm relies on unreasonable parameter settings. In particular, the algorithm set a threshold $\sigma$, and requires that when each node pushes residues to its neighbors, the total amount of residues pushed along each edge is at most $\sigma$. As a consequence, if a node $v$ has $d$ edges, then the total amount of residues pushed from $v$ is upper bounded by $d \cdot \sigma$. Therefore, if the initial residue of $v$ is $x$, then the approximate PPR for $v$ returned by the algorithm is at most $x - d \cdot \sigma$. In the experiments, we have $x = 1$ for each initial node and $\sigma = 10^{-6}$, and the average degree of each node is only a few dozens. As a result, the approximate PPR for each initial node is almost identical to the initial residue $x = 1$. Apparently, such approximate PPR values are far from the ground truth PPR values. This issue cannot be addressed by using a larger $\sigma$, because the amount of noise used by the algorithm is linear to $\sigma$.

Third, the paper presents algorithms for both edge DP and joint edge DP, but the experiments only evaluate the joint edge DP algorithms. As a consequence, it is unclear whether the proposed edge DP algorithms work well or not.

---

> ### Author Response · Authors · 2022-08-02
> **Response to Reviewer MDNv**
>
>  Thank you for the review and comments. In the following, we address your concerns.
>
> -"the proposed PPR algorithm is relatively straightforward, it computes the PPR values and then adds noise based on the sensitivity of the values": Notice that our algorithm did not compute the standard PPR. Our main technical contribution is the design of the "capped PushFlow" algorithm which we have introduced explicitly to show that its sensitivity can be always bounded by a parameter sigma and thus allowing us to add limited amounts of noise. This innovation is critical as using the “standard PPR” + noise would require adding so much noise that the signal is lost. This is something that we have verified also experimentally. We tried adding Laplace noise using the sensitivity bound (which is 1) for the standard PPR vector. The performance of this baseline was much poorer than the  randomized response baseline and close to random so we omitted it. We will mention it in the final version of the paper.
>
> -”The sensitivity analysis is mostly based on the triangle inequality and is rather loose”: Our sensitivity analysis is actually provably tight. According to Corollary 4.6, our approximation guarantee holds when the sensitivity parameter >= 1/D^2 for joint edge DP and >= 1/D for edge DP where D is the minimum degree of the graph. This matches the lower bound (we also mentioned this in section 1.1) shown in Appendix A (see the supplementary material). In particular we show that the ground truth PPR has sensitivity Omega(1/D) and joint sensitivity Omega(1/D^2). Our results are actually theoretically optimal up to constant factors.
>
> -”As a result, the approximate PPR for each initial node is almost identical to the initial residue  x=1”: Actually it is possible to observe theoretically and empirically that the residual is pushed in great amounts and not at all similar to the initial residual (x=1). For instance for joint edge DP, sigma can be much smaller than edge DP as shown by our theoretical lower bounds. In addition, for joint edge DP, the threshold of the source is set to be infinity, and thus it can push arbitrarily large flow from the source (see line 3 of Algorithm 2, item 1). We have conducted an additional experiment to confirm that the residual is pushed out of the source in all realistic settings of the experiments. For that, we have logged the remaining residual for the initial node across the sensitivity parameter for the datasets used in the experiments. In the case of joint edge DP used in the experiments, we observe that the average residual is less than 1e-6 across the sensitivity parameter, thus, almost all of the residual is being pushed from the source node.
>
> -”the experiments only evaluate the joint edge DP algorithm”: that is correct. As our lower bound shows, there is a much stronger lower bound for sensitivity for edge DP (see Appendix A in the supplementary material), i.e., the ground truth PPR has Omega(1/D) sensitivity. Due to large sensitivity, the practical performance of our edge DP version (while almost optimal in theory) is less good than the joint edge DP version. In contrast, the theoretical lower bound of joint edge level DP is only Omega(1/D^2), and we show its practical performance is very good. Since the joint-dp setting is the most practically relevant one for personalization (as in PPR) we focus on these settings for our experiments.

---

> > ### Author Response · Authors · 2022-08-10
> > **feedback on the responses**
> >
> > Thanks again for your comments. We think we addressed all your major concerns in our response. Would be great if you can comment on our responses. Thanks.

---

### Author Response · Authors · 2022-08-09
**Discussion period closing today**

We would like to thank all the reviewers again for their comments.

As the discussion period ends today, we would like gently to request the reviewers to respond to our comments with their feedback before the deadline. This way we can make sure that, if any questions arise, we are able to provide all the clarifications needed in time.

Thank you for your work.
Best regards,

---

### Meta-Review · Area_Chair_X3v1 · 2022-08-24

**Recommendation:** Accept
**Confidence:** Less certain

**Metareview:**

Paper studies computing PPR in differential privacy setting. Given the importance of PPR in real world applications, we recommend accepting the paper as it brings an important problem to the DP community. However, we encourage authors to incorporate the comments from the authors, make sure that all the details of the proofs are made available in the final version, and clarify any comments reviewers raised. In particular, we encourage the authors to adequately address the criticisms of @Reviewer MDNv in the true spirit of science.

**Award:**

No

---

### Decision · Program_Chairs · 2022-09-14

Accept